# Seeing It Before It Happens: In-Generation NSFW Detection for Diffusion-Based Text-to-Image Models

## Abstract

Diffusion-based text-to-image (T2I) models enable high-quality image generation but also pose significant risks of misuse, particularly in producing not-safe-for-work (NSFW) content. While prior detection methods have focused on filtering prompts before generation or moderating images afterward, the in-generation phase of diffusion models remains largely unexplored for NSFW detection. In this paper, we introduce In-Generation Detection (IGD), a simple yet effective approach that leverages the predicted noise during the diffusion process as an internal signal to identify NSFW content. This approach is motivated by preliminary findings suggesting that the predicted noise may capture semantic cues that differentiate NSFW from benign prompts, even when the prompts are adversarially crafted. Experiments conducted on seven NSFW categories show that IGD achieves an average detection accuracy of 92.45% over naive and adversarial NSFW prompts, outperforming seven baseline methods.

## 1 Introduction

Text-to-image (T2I) models, powered by diffusion architectures, now generate high-quality images from natural language prompts. Systems like Stable Diffusion Rombach et al. (2022), DALL·E Ramesh et al. (2022), and SDXL Podell et al. (2023) are widely adopted in design, content creation, and virtual environments. However, their capabilities also pose risks Liu et al. (2024b); Zhang et al. (2024), especially misuse for illegal not-safe-for-work (NSFW) content. While some NSFW prompts are explicit and easily flagged, others are adversarially crafted to evade filters by manipulating language or exploiting model weaknesses Yang et al. (2024c;a); Tsai et al. (2024); Chin et al. (2023). These challenges underscore the need for stronger NSFW detection.

Existing NSFW detection methods for T2I models can be broadly categorized into two types (see red boxes in Figure 1): pre-detection, which analyzes prompts before generation, and post-detection, which evaluates the final image Liu et al. (2024b). Pre-detection methods rely on lexical or classifier-based filters Li (2023); Hanu & Unitary team (2020), but are vulnerable to adversarially crafted prompts that use misspellings or vague language to obfuscate intent Yang et al. (2024a); Tsai et al. (2024). Post-detection offers more accurate results CompVis (2022a); OpenAI (2025); Aliyun (2025); Azure (2025), but introduces detection latency and more resource consumption Zhang et al. (2024).

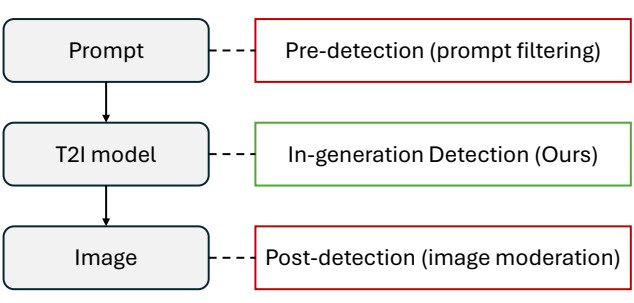

Figure 1: Overview of different NSFW detection types.

While existing methods primarily focus on pre-detection (prompt filtering) and post-detection (image moderation), the possibility of detecting NSFW content during the image generation process itself has, to our knowledge, been largely overlooked. In diffusion-based T2I models, generation unfolds

gradually over a sequence of denoising steps, offering a rich intermediate space that could be monitored in real time. However, this in-generation phase remains underexplored for NSFW detection.

To this end, we propose In-Generation Detection (IGD), a simple yet effective method for identifying NSFW intent during image generation in diffusion models. IGD monitors predicted noise across denoising steps, reflecting the evolving semantics of the prompt. Leveraging this signal, we train a lightweight classifier to detect whether the given prompt is intended to produce NSFW content. The module integrates into the generation loop, enabling early intervention before full synthesis. Empirical observations show predicted noise serves as a highly discriminative feature for NSFW detection, capturing semantic differences between NSFW and SFW prompts. This also holds for adversarial prompts crafted to prompt filtering. Despite their obfuscated surface form, the predicted noise patterns resemble naive NSFW prompts, as both share the intent to generate NSFW content. In contrast, SFW prompts tend to produce distinct noise patterns, enabling IGD to separate both naive and adversarial NSFW prompts from benign ones during generation. To sum up, IGD is lightweight, easy to integrate into T2I models, enables early intervention, and remains robust against adversarial prompts, making it a practical solution for safer diffusion-based T2I systems. Experiments on seven NSFW categories show that IGD achieves an average detection accuracy of 92.45% across both naive and adversarial prompts, outperforming seven baseline methods.

In summary, our key contributions are:

- To the best of our knowledge, this is the first NSFW detection method that operates *in-generation* by leveraging intermediate representations from diffusion-based T2I models, introducing a third paradigm beyond prompt filtering and image moderation.
- Motivated by the observation that predicted noise in diffusion models encodes semantically discriminative patterns, we propose IGD, a simple yet effective method that leverages this signal for in-generation detection.
- Experiments on seven NSFW categories show that IGD outperforms seven baselines and remains effective against adversarial prompts from five attack methods.

## 2 RELATED WORK

### 2.1 NSFW GENERATION

Prior work on NSFW generation in T2I models falls into two approaches. The first collects explicit prompts from online forums or NSFW communities (e.g., I2P Schramowski et al. (2023a)), termed **naive NSFW prompts**. The second focuses on adversarial prompting, where seemingly benign inputs are crafted to evade safety filters while still triggering NSFW outputs, termed **adversarial NSFW prompt**. Examples include SneakyPrompt Yang et al. (2024c), which applies reinforcement learning to inject subtle perturbations. MMA-Diffusion Yang et al. (2024a), which generates multimodal noise to mislead encoders. Ring-A-Bell Tsai et al. (2024), embedding semantic residues into innocuous prompts. P4D Chin et al. (2023), using automated red-teaming to expose vulnerabilities. DiffZOO Dang et al. (2024), enabling query-based black-box attacks via zeroth-order optimization.

### 2.2 NSFW DETECTION

Most existing NSFW detection approaches in T2I models operate at two stages: **pre-detection** (prompt-level) and **post-detection** (image-level).

Pre-detection methods analyze the input prompt to assess potential risks before image generation, from simple keyword filters to classifiers like NSFW-text-classifier Li (2023) and Detoxify Hanu & Unitary team (2020). Yet they remain vulnerable to adversarial tricks (e.g., synonyms, Unicode, incoherent phrasing). Embedding-based approaches such as GuardT2I Yang et al. (2024b) and Latent Guard Liu et al. (2024a) improve robustness but still falter under surface-level perturbations.

Post-detection methods evaluate the generated image to determine whether it contains NSFW content. The Safety Checker CompVis (2022a) in Stable Diffusion uses CLIP-based classifiers to flag sensitive visuals, while services like OpenAI Moderation API OpenAI (2025), Aliyun Aliyun (2025), and Azure Azure (2025) provide multimodal screening across harmful categories. Though effective, these approaches add latency and only act after generation, wasting resources on unsafe outputs.

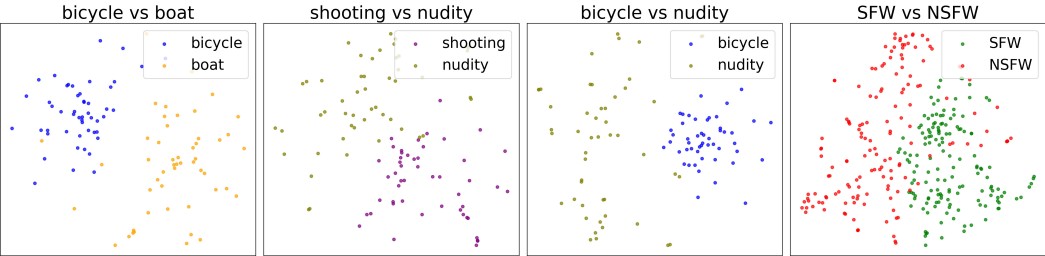

Figure 2: t-SNE visualizations of predicted noise $\epsilon_t$ across four representative category pairs.

## 3  PRELIMINARY AND MOTIVATION

### 3.1  PRELIMINARY

**Diffusion models** are a type of generative model Ho et al. (2020) that decomposes the data generation process into two complementary stages: a forward process and a reverse process. In the forward process, noise is gradually added to the input image, transforming the original data distribution into a standard Gaussian noise. Conversely, the reverse process is trained to recover the original data from pure noise by learning to invert the corruption process.

Given an input image latent $x^0$, the forward process perturbs the data using a predefined noise schedule $\{\beta^t : \beta^t \in (0,1)\}_{t=1}^T$, which controls the magnitude of noise added over $T$ steps. This results in a sequence of noisy latent variables $\{x^1, x^2, \ldots, x^T\}$. At each timestep $t$, the noisy sample $x^t$ is generated as:

$$x^t = \sqrt{\bar{\alpha}^t}x^0 + \sqrt{1 - \bar{\alpha}^t}\,\epsilon, \tag{1}$$

where $\alpha^t = 1 - \beta^t$, $\bar{\alpha}^t = \prod_{s=1}^t \alpha^s$, and $\epsilon \sim \mathcal{N}(0, \mathbf{I})$ represents standard Gaussian noise.

The reverse process aims to denoise $x^{t+1}$ to obtain a less noisy $x^t$ by estimating the noise component $\epsilon$ using a neural network $\epsilon_\theta(x^{t+1}, t)$. The model is trained to minimize the $\ell_2$ distance between the true noise and the predicted noise:

$$\mathcal{L}_{\text{uncondition}} = \mathbb{E}_{x_0, t, \epsilon \sim \mathcal{N}(0,1)} \left\| \epsilon - \epsilon_\theta(x^{t+1}, t) \right\|_2^2, \tag{2}$$

where $t$ is uniformly sampled from $\{1, \ldots, T\}$.

In contrast to unconditional diffusion models, conditional (prompt-based) diffusion models guide the generation process using an additional condition or prompt $c$. This enables the model to produce photorealistic outputs that are semantically aligned with the given text prompt or concept. The training objective is then extended as:

$$\mathcal{L}_{\text{cond}} = \mathbb{E}_{x_0, t, c, \epsilon \sim \mathcal{N}(0,1)} \left\| \epsilon - \epsilon_\theta(x^{t+1}, t, c) \right\|_2^2. \tag{3}$$

### 3.2  MOTIVATION

According to Eq. (3), the reverse process of diffusion (i.e., the denoising procedure) predicts the noise at timestep $t$ as:

$$\epsilon_t = \epsilon_\theta(x^{t+1}, t, c), \tag{4}$$

where $x^{t+1}$ is the current noisy latent, $t$ is the diffusion timestep, and $c$ is the embedding of the text condition. Since the model is trained to generate images conditioned on $c$ (the embedding of the input prompt), the predicted noise $\epsilon_t$ naturally becomes a condition-dependent variable that reflects the semantic intent of the prompt and implicitly encodes information about the generated content. Consequently, it has the potential to serve as an effective feature for distinguishing the intent of NSFW and SFW prompts. **However, the potential of predicted noise in discriminative tasks, particularly NSFW detection, remains unexplored.**

**Predicted noise reflects separable generation intents for NSFW and SFW prompts.** To explore whether the predicted noise $\epsilon_t$ encodes meaningful semantic information, we conduct t-SNE Maaten

& Hinton (2008) visualizations using $\epsilon_t$ extracted by Stable Diffusion v1.5 CompVis (2022c) from prompts associated with different semantic labels. To be specific, we randomly select three SFW categories, including *bicycle*, *boat*, and *train*, as well as three NSFW categories, including *bloody*, *nudity*, and *shooting*. For each category, we use 50 different input prompts and analyze the distribution of the corresponding predicted noise $\epsilon_t$ at a random diffusion timestep. We also aggregate these into two broader classes, SFW and NSFW, to visualize their overall separation. As shown in Figure 2, we illustrate comparisons between: ❶ two SFW categories (*bicycle* vs. *boat*); ❷ two NSFW categories (*shooting* vs. *nudity*); ❸ a direct SFW vs. NSFW category comparison (*bicycle* vs. *nudity*); and ❹ an aggregated comparison of three SFW categories (*bicycle*, *boat*, *train*) and three NSFW categories (*bloody*, *nudity*, *shooting*) (See Appendix A.2 for more details). The resulting t-SNE plots show that the predicted noise forms semantically coherent clusters across different categories, with SFW and NSFW classes largely separated in embedding space. This indicates that $\epsilon_t$ encodes meaningful semantic structure before image synthesis is complete. While the analysis does not cover the full range of prompts, it offers important empirical insight: predicted noise captures discriminative patterns and holds strong potential as an in-generation signal for NSFW detection.

**Predicted noise as a signal for detecting adversarial NSFW prompts.** Adversarially crafted NSFW prompts can evade prompt filtering by subtly modifying surface text while preserving semantic intent, exposing a key weakness of pre-detection methods relying on prompt embeddings. In contrast, our in-generation approach IGD remains effective because both naive and adversarial NSFW prompts drive the model to generate unsafe content. This shared intent is reflected in the predicted noise during generation, which remains similar across prompt types despite textual obfuscation. Leveraging this convergence, IGD achieves greater robustness to adversarial prompts than pre-detection methods.

To illustrate this, we compare prompt embeddings and predicted noise features using representations from Stable Diffusion v1.5. Specifically, we use the *sexual* category from the I2P dataset as naive NSFW prompts and generate adversarial variants using the Ring-A-Bell attack. For each prompt, we extract (1) the prompt embedding from the text encoder and (2) the predicted noise from the diffusion process. As shown in Figure 3, the t-SNE visualization of prompt embeddings (left) shows a clear separation between naive and adversarial prompts. This reflects the success of adversarial attacks in manipulating surface text to shift the prompt embedding distribution. Since prompt-level classifiers typically rely on these embeddings to estimate NSFW likelihood, such separation allows adversarial prompts to evade detection despite

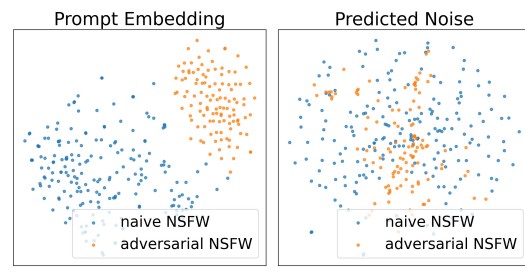

Figure 3: Compare prompt embedding and predicted noise between naive&adv NSFW prompt.

preserving the original NSFW intent. In contrast, the predicted noise features (right) exhibit strong overlap between naive and adversarial prompts, suggesting that both lead to similar generative behavior during denoising. This convergence reveals that the underlying visual intent remains consistent, even if the textual form is obfuscated.

These observations suggest a promising direction: since predicted noise shows separable patterns between SFW and NSFW prompts, as well as consistency across naive and adversarial NSFW prompts, it may serve as a useful feature for classification. Motivated by this, we design a classifier that leverages predicted noise during generation for NSFW detection. Our experiments later indicate that this approach can provide good results.

## 4 METHOD

### 4.1 PROBLEM FORMULATION

The goal of NSFW detection is to determine whether a given input prompt $p$ generates inappropriate or sensitive visual content. Formally, let $p$ denote a text prompt, and let $y \in \{0, 1\}$ be the corresponding label, where $y = 1$ indicates that the image generated from $p$ contains NSFW content, and $y = 0$ otherwise. The task is to learn a function $f(p) \rightarrow y$ that predicts the safety label of the image

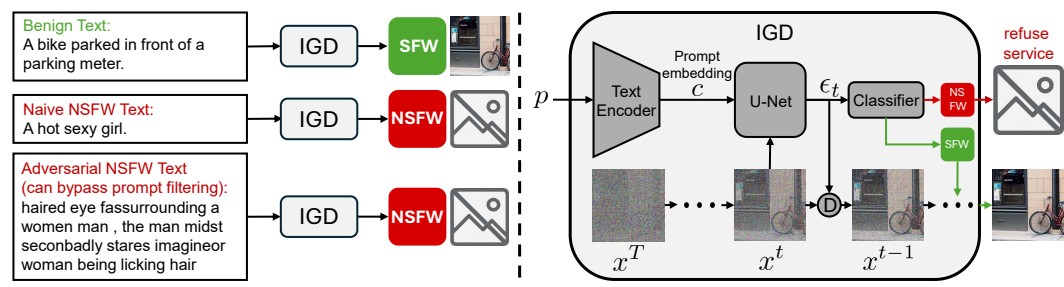

Figure 4: The overall framework of IGD.

conditioned on the prompt. Although $y$ reflects the semantic class of the final image, the prediction can be based on various forms of input, such as the prompt itself, intermediate features during generation, or the final image. The output $y$ serves as a binary decision signal for content moderation.

## 4.2 IN-GENERATION DETECTION METHOD

We propose IGD, a simple yet effective in-generation NSFW detection method, as illustrated in Figure 4. Unlike prior approaches that perform classification on the input prompt or output image, IGD leverages intermediate predicted noise from the diffusion process to identify unsafe content before image synthesis completes.

As illustrated in the right part of Figure 4, given an input text prompt $p$, we obtain its embedding $c$ using a text encoder $E_{\text{text}}$: $c = E_{\text{text}}(p)$. During the denoising of a diffusion model (e.g., Stable Diffusion), the U-Net denoiser $\epsilon_\theta$ predicts the noise at each timestep $t$ as $\epsilon_t = \epsilon_\theta(x^{t+1}, t, c)$. We then attach a lightweight binary classifier $f_\phi(\cdot)$ to the predicted noise $\epsilon_t$ and define the NSFW decision as:

$$y = f_\phi(\epsilon_t), \tag{5}$$

where $y \in \{0, 1\}$ indicates whether the predicted image is classified as NSFW. If $y = 1$, the generation is terminated early to prevent the synthesis of unsafe image. Otherwise, the process continues as usual. As illustrated in the left part of Figure 4, IGD effectively handles benign prompts, naive NSFW descriptions, and even adversarially obfuscated texts that bypass prompt-level filters.

This method offers several benefits. ❶ Compared to pre-detection (prompt filtering), IGD is more *robust* to obfuscated prompts. While prompt-based classifiers rely on surface text that can be easily manipulated, IGD analyzes the predicted noise $\epsilon_t$, which reflects how the model internally interprets and visualizes the prompt. This makes it less sensitive to minor textual variations and better aligned with the actual generation intent. ❷ Compared to post-detection (image moderation), IGD enables *early intervention* by analyzing internal generative signals before image synthesis completes. ❸ Finally, the classifier is lightweight, consisting of a small number of neural layers, and introduces negligible overhead to the generation process. In fact, IGD is highly efficient, adding only 0.0044s of inference time compared to the 5.304s required by the Stable Diffusion process on an NVIDIA RTX 3090 GPU. Notably, this makes IGD considerably faster than post-detection methods, which must wait until the entire generation process has finished before performing classification.

## 5 EXPERIMENT

### 5.1 EXPERIMENTAL SETUPS

**Datasets.** We conducted experiments on two datasets to train our NSFW detector: **I2P** Schramowski et al. (2023a) and **MSCOCO** Lin et al. (2014). The I2P dataset contains 4,703 manually crafted NSFW prompts targeting T2I models, covering seven NSFW categories: self-harm, violence, shocking content, hate, harassment, sexual, and illegal activity. We sample 200 prompts from each category, resulting in 1,400 NSFW training examples. For clean data, we use the training split of COCO2014, which includes 123,287 images, each paired with five human-written captions. We extract the first

caption from each image and randomly sample 1,400 clean prompts. In total, the training set consists of 2,800 samples, evenly balanced between NSFW and clean data.

**Baselines.** We compare our method with **seven** representative moderation tools, covering open-source models, commercial APIs, and T2I-specific defenses. **NSFW-text-classifier** Li (2023) is a Hugging Face-hosted binary classifier for NSFW text detection. **Detoxify** Hanu & Unitary team (2020) is a transformer-based toxicity detector trained on Jigsaw's dataset. **OpenAI Moderation API** Markov et al. (2023); OpenAI (2025), **Aliyun Text Moderation** Aliyun (2025), and **Azure AI Content Safety** Azure (2025) are commercial services supporting text inputs. They detect harmful content categories such as sexual, violent, political, and hate-related material, using advanced multimodal or multilingual models. **Latent Guard** Liu et al. (2024a) detects adversarial prompts in the latent space of T2I embeddings without retraining. **GuardT2I** Yang et al. (2024b) interprets prompt embeddings via a conditional language model to identify harmful intent while preserving generation quality.

**Target model.** Following the same setting as previous baselines Latent Guard Liu et al. (2024a) and GuardT2I Yang et al. (2024b), we adopt Stable Diffusion v1.5 Rombach et al. (2022); CompVis (2022c), a widely used open-source T2I model, as our target model for obtaining predicted noise.

**Evaluation metrics.** Following prior works Liu et al. (2024a); Yang et al. (2024b), we adopt three metrics: **Accuracy**, **AUROC**, and **FPR@TPR95**. Accuracy measures classification correctness. AUROC reflects discriminative ability across thresholds. FPR@TPR95 evaluates robustness under high recall. Higher accuracy and AUROC, and lower FPR@TPR95, indicate better performance.

**Implementation details.** We use Stable Diffusion v1.5 as the target T2I model to extract the predicted noise as the feature for classification. The total number of inference steps of the T2I model is 50, and the timestep we used for predicted noise extraction is 5. We employ a straightforward 5-layer fully connected MLP as a binary classifier, trained using the size-unfolded predicted noise as input features. Importantly, this classifier is trained solely on the naive NSFW and clean prompts from our constructed training set, without access to adversarial or paraphrased examples. The model is optimized using the Adam optimizer with a learning rate of $1e^{-3}$ for 100 epochs. We conduct our experiments on an NVIDIA RTX 3090 GPU with 24GB of memory.

## 5.2 COMPARISON TO BASELINES

We evaluate our method against seven representative NSFW defense methods. The evaluation is conducted on the I2P dataset, which includes seven NSFW categories: sexual, violence, self-harm, harassment, hate, shocking, and illegal activity. For each category, we sample 100 prompts to construct the naive NSFW prompt dataset, except for the hate category, which contains only 47 available prompts, resulting in a total of 647 prompts for evaluation. To ensure the validity of evaluation metrics (e.g., accuracy), we pair each NSFW prompt with a clean (*i.e.*, SFW) prompt sampled from the MSCOCO 2014 training set. This procedure results in a clean prompt set that matches the size of each corresponding naive NSFW prompt set. In total, 647 clean prompts are sampled. This is the naive NSFW dataset.

Table 1: Compare with baselines.

|  | Methods | Naive | Adversarial | Average |
|---|---|---|---|---|
| Accuracy ↑ | NSFW-text-classifier | 58.81% | 71.12% | 64.97% |
|  | Detoxify | 50.54% | 56.38% | 53.46% |
|  | OpenAI Moderation API | 57.50% | 66.07% | 61.78% |
|  | Aliyun Text Moderation | 52.78% | 56.93% | 54.86% |
|  | Azure AI Content Safety | 56.96% | 72.77% | 64.86% |
|  | Latent Guard | 57.26% | 61.29% | 59.28% |
|  | GuardT2I | 51.70% | 65.20% | 58.45% |
|  | IGD (Ours) | **90.96%** | **93.94%** | **92.45%** |
| AUROC ↑ | NSFW-text-classifier | 58.65% | 62.76% | 60.71% |
|  | Detoxify | 56.60% | 71.56% | 64.08% |
|  | OpenAI Moderation API | 88.88% | 93.17% | 91.03% |
|  | Aliyun Text Moderation | 52.78% | 56.93% | 54.86% |
|  | Azure AI Content Safety | 53.44% | 74.52% | 63.98% |
|  | Latent Guard | 66.91% | 70.51% | 68.71% |
|  | GuardT2I | 87.17% | 96.88% | 92.03% |
|  | IGD (Ours) | **95.48%** | **98.07%** | **96.78%** |
| FPR@TPR95 ↓ | NSFW-text-classifier | 90.73% | 90.45% | 90.59% |
|  | Detoxify | 98.30% | 90.36% | 94.33% |
|  | OpenAI Moderation API | 44.51% | 34.71% | 39.61% |
|  | Aliyun Text Moderation | 100.00% | 100.00% | 100.00% |
|  | Azure AI Content Safety | 99.54% | 97.61% | 98.57% |
|  | Latent Guard | 85.63% | 79.16% | 82.39% |
|  | GuardT2I | 56.88% | 15.06% | 35.97% |
|  | IGD (Ours) | **26.12%** | **7.44%** | **16.78%** |

total, 647 clean prompts are sampled. This is the naive NSFW dataset.

To assess the robustness of our IGD method under adversarial conditions, we construct the adversarial NSFW prompt dataset independent of the naive NSFW dataset. Specifically, we sample 200 prompts from the sexual category of the I2P dataset (denoted as I2P-sexual, which is entirely distinct from those used in the naive NSFW dataset), and apply five state-of-the-art attack methods to automatically generate adversarial prompts. We focus on the sexual category as it is the only NSFW type consistently

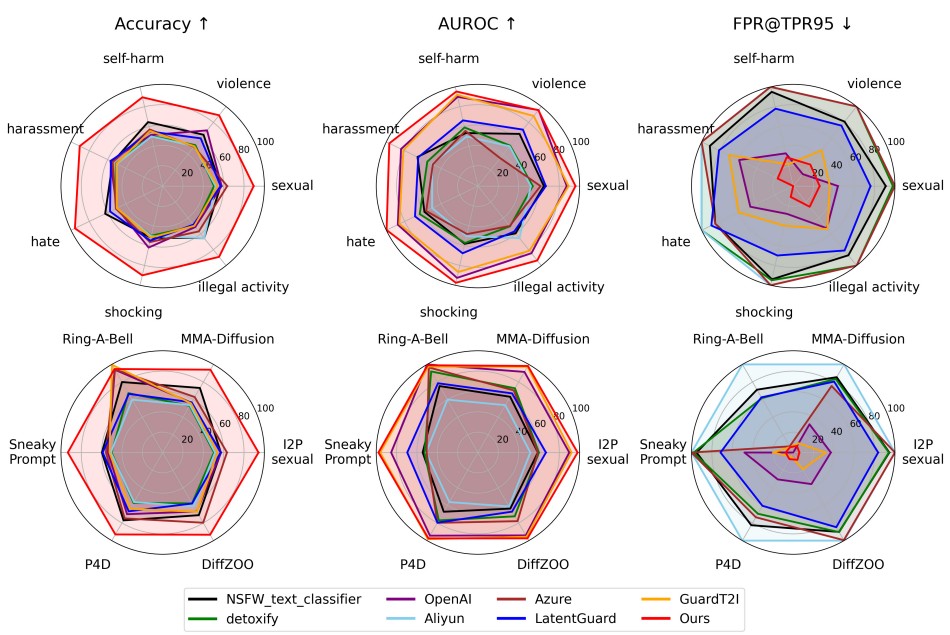

Figure 5: Radar plots of our method and baselines against various adversarial NSFW prompts.

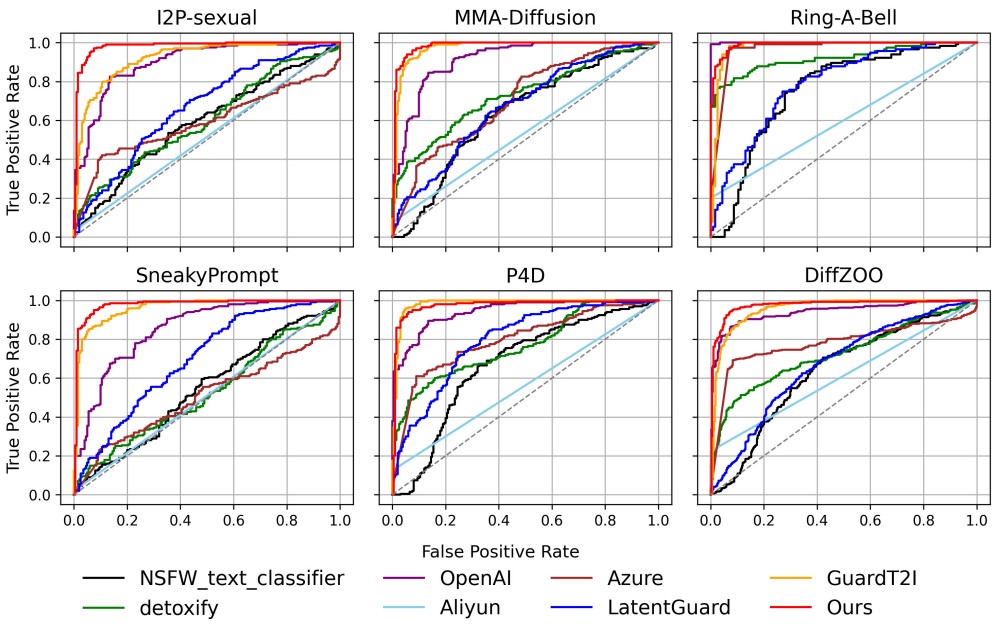

Figure 6: ROC curves of our method and baselines against various adversarial NSFW prompts.

supported by all evaluated defense methods, ensuring a fair and comparable adversarial evaluation. Following the same evaluation protocol as in the naive setting, we sample an equal number of clean prompts from the MSCOCO 2014 training set to construct a balanced clean prompt set for adversarial evaluation. Together, the adversarial NSFW prompts and the sampled clean prompts form our final adversarial NSFW prompt dataset.

As shown in Table 1, IGD achieves the best performance on both naive and adversarial NSFW prompt datasets, with 92.45% accuracy, 96.78% AUROC, and 16.78% FPR@TPR95. The results demonstrate its effectiveness in distinguishing NSFW content while remaining robust under adversarial conditions.

Table 2: Comparison with concept-erasing methods on naive NSFW prompt.

| Methods | Naive NSFW Prompt | | | | | | | |
|---|---|---|---|---|---|---|---|---|
| | sexual | violence | self-harm | harassment | hate | shocking | illegal activity | Average |
| ESD | 76.00% | 97.00% | 88.00% | 98.00% | 100.00% | 89.00% | 99.00% | 92.43% |
| SLD-weak | 42.00% | 92.00% | 70.00% | 92.00% | 97.87% | 75.00% | 99.00% | 81.12% |
| SLD-strong | 58.00% | 96.00% | 81.00% | 90.00% | 93.62% | 82.00% | 96.00% | 85.23% |
| IGD (Ours) | **12.00%** | **13.00%** | **12.00%** | **10.00%** | **2.13%** | **11.00%** | **13.00%** | **10.45%** |

Table 3: Comparison with concept-erasing methods on adversarial NSFW prompt.

| Methods | Adversarial NSFW Prompt | | | | | | |
|---|---|---|---|---|---|---|---|
| | I2P-sexual | MMA-Diffusion | Ring-A-Bell | SneakyPrompt | P4D | DiffZOO | Average |
| ESD | 76.00% | 85.00% | 71.30% | 87.00% | 55.00% | 87.97% | 77.05% |
| SLD-weak | 48.00% | 45.50% | 95.65% | 36.50% | 86.00% | 56.42% | 61.34% |
| SLD-strong | 35.50% | 36.50% | 92.17% | 29.00% | 79.00% | 48.13% | 53.38% |
| IGD (Ours) | **2.50%** | **4.50%** | **3.00%** | **1.74%** | **5.00%** | **6.42%** | **3.86%** |

**Comparison with baselines on detailed NSFW types.** As illustrated in Figure 5, the first row shows results on naive NSFW prompts, with the three subplots from left to right corresponding to Accuracy, AUROC, and FPR@TPR95. The second row shows the same set of metrics under adversarial NSFW prompts. This layout provides a direct visual comparison of how different methods behave across both naive and adversarial settings. IGD consistently maintains superior performance under these challenging adversarial settings, with accuracy and AUROC remaining at high levels and FPR@TPR95 keeping at the lowest among all compared methods. These radar plots collectively demonstrate the fine-grained discrimination capacity of IGD, confirming its effectiveness against both naive and adversarial NSFW prompts. (See Appendix A.3 for more detailed data).

In Figure 6, we present the ROC curves of various baselines alongside our proposed IGD for comparison. Each subfigure represents an adversarial NSFW prompt attack scenario. IGD exhibits consistently strong discriminative capability, achieving the best ROC curves in I2P-sexual, MMA-Diffusion, SneakyPrompt, and DiffZOO. In these scenarios, IGD maintains a sharp rise in the ROC curve, underscoring its robustness against adversarial NSFW prompts.

Overall, these results show IGD achieves high detection accuracy, maintains low false positive rates under strict recall, and defense effectively against both naive and adversarial NSFW prompts.

## 5.3 COMPARISON WITH CONCEPT ERASING METHODS

Unlike traditional NSFW defense methods that rely on classification or detection to identify harmful prompts, concept-erasing approaches work by removing specific concepts from the model itself. Although not originally intended for NSFW defense, they can also reduce the generation of NSFW content. Therefore, we include them in our comparisons as complementary references.

ESD Gandikota et al. (2023) and SLD Schramowski et al. (2023b) are representative concept-erasing approaches. Unlike classification-based methods, these models do not produce explicit classification outputs. Following GuardT2I, we assess their effectiveness using the Attack Success Rate (ASR), which is determined by applying NudeNet Bedapudi (2019) to evaluate whether generated images contain nudity. A lower ASR indicates a more effective defense. For both ESD and SLD, we use the publicly released checkpoints from their official implementations, which have been fine-tuned to erase the concept of "nudity". Following the SLD paper, we adopt both the weak and strong settings, referred to as SLD-weak and SLD-strong, respectively. All baseline models used in our evaluation are obtained directly from the official releases of the original papers.

As shown in Table 2 and Table 3, in terms of ASR, our method achieves an average of 10.45% and 3.86%, significantly lower than all concept-erasing baselines. This substantial reduction demonstrates our method outperforms the concept-erasing method in the NSFW detection task.

Table 4: Comparison on different timesteps.

| Timestep | 5 | 10 | 15 | 20 | 25 | 30 | 35 | 40 | 45 | 50 | Average |
|---|---|---|---|---|---|---|---|---|---|---|---|
| Accuracy ↑ | 90.96% | 84.47% | 80.76% | 83.62% | 84.00% | 85.01% | 87.64% | 88.18% | 89.80% | 90.26% | 86.47% |
| AUROC ↑ | 95.48% | 92.05% | 88.28% | 91.21% | 91.68% | 92.96% | 94.60% | 95.33% | 95.89% | 96.16% | 93.36% |
| FPR@TPR95 ↓ | 26.12% | 35.09% | 53.63% | 42.35% | 42.19% | 35.70% | 30.14% | 26.43% | 25.04% | 20.56% | 33.72% |

Table 5: Performance of IGD with different target models on detecting naive NSFW prompts.

| | Methods | Naive NSFW Prompt | | | | | | | |
|---|---|---|---|---|---|---|---|---|---|
| | | sexual | violence | self-harm | harassment | hate | shocking | illegal activity | Average |
| Accuracy ↑ | Stable Diffusion v1.4 | 90.00% | 89.50% | 91.50% | 90.00% | 94.68% | 91.00% | 84.00% | 90.10% |
| | Stable Diffusion v1.5 | 89.50% | 89.00% | 89.50% | 90.50% | 95.74% | 90.00% | 89.00% | 90.46% |
| | Stable Diffusion v2.1 | 91.00% | 90.00% | 90.50% | 91.50% | 87.23% | 91.50% | 85.00% | 89.53% |
| AUROC ↑ | Stable Diffusion v1.4 | 95.58% | 96.91% | 97.93% | 96.30% | 98.87% | 97.44% | 90.92% | 96.28% |
| | Stable Diffusion v1.5 | 95.92% | 95.35% | 95.34% | 96.75% | 99.68% | 97.52% | 93.55% | 96.30% |
| | Stable Diffusion v2.1 | 97.03% | 96.41% | 97.92% | 96.89% | 96.02% | 97.66% | 93.13% | 96.44% |
| FPR@TPR95 ↓ | Stable Diffusion v1.4 | 17.00% | 17.00% | 11.00% | 21.00% | 2.13% | 11.00% | 58.00% | 19.59% |
| | Stable Diffusion v1.5 | 26.00% | 27.00% | 29.00% | 17.00% | 0.00% | 11.00% | 26.00% | 19.43% |
| | Stable Diffusion v2.1 | 12.00% | 13.00% | 15.00% | 13.00% | 29.79% | 11.00% | 35.00% | 18.40% |

Table 6: Performance of IGD with different target models on detecting adversarial NSFW prompts.

| | Methods | Adversarial NSFW Prompt | | | | | | |
|---|---|---|---|---|---|---|---|---|
| | | I2P-sexual | MMA-Diffusion | Ring-A-Bell | SneakyPrompt | P4D | DiffZOO | Average |
| Accuracy ↑ | Stable Diffusion v1.4 | 94.00% | 94.00% | 94.35% | 94.25% | 93.00% | 93.05% | 93.77% |
| | Stable Diffusion v1.5 | 94.25% | 94.00% | 94.78% | 93.25% | 93.00% | 93.32% | 93.77% |
| | Stable Diffusion v2.1 | 91.50% | 91.50% | 90.87% | 90.75% | 91.50% | 91.84% | 91.33% |
| AUROC ↑ | Stable Diffusion v1.4 | 98.71% | 98.81% | 99.15% | 98.60% | 98.17% | 98.19% | 98.60% |
| | Stable Diffusion v1.5 | 98.00% | 98.25% | 99.02% | 97.81% | 97.45% | 97.44% | 98.00% |
| | Stable Diffusion v2.1 | 97.73% | 97.81% | 96.85% | 97.57% | 96.96% | 98.15% | 97.51% |
| FPR@TPR95 ↓ | Stable Diffusion v1.4 | 2.00% | 4.00% | 2.61% | 2.50% | 3.00% | 9.09% | 3.87% |
| | Stable Diffusion v1.5 | 6.00% | 7.50% | 6.09% | 7.50% | 7.50% | 8.29% | 7.15% |
| | Stable Diffusion v2.1 | 9.50% | 7.00% | 22.61% | 13.00% | 11.00% | 11.76% | 12.48% |

## 5.4 DISCUSSION

**Timestep selection.** To assess the effect of timestep on NSFW detection, we evaluate the discriminative power of predicted noise extracted at different stages of the diffusion process on naive NSFW prompt set. As shown in Table 4, predicted noise across timesteps consistently shows strong performance, confirming the stability of our in-generation signal. While later timesteps (e.g., 50) offer slightly higher accuracy, early detection is preferred for efficiency and timely intervention. We therefore adopt timestep 5 as a practical choice while enabling response before image synthesis.

**Different T2I models.** To evaluate cross-model generalizability, we apply IGD to features from different diffusion-based T2I models Rombach et al. (2022): Stable Diffusion v1.4 CompVis (2022b), v1.5 CompVis (2022c), and v2.1 CompVis (2022d). In Table 5 and Table 6, IGD consistently achieves high accuracy and AUROC, along with a low FPR@TPR95 on average across naive and adversarial NSFW prompts, confirming robustness and effectiveness regardless of the underlying T2I model.

**More Experiments.** We further evaluate our method on two additional datasets (4chan Qu et al. (2023) and Lexica Qu et al. (2023)) in Appendix A.4, examining detection performance with extended timesteps in Appendix A.5, the effect of varying classifier depth in Appendix A.6, and the feasibility of multi-class classification on the categories of the I2P dataset in Appendix A.7.

## 6 CONCLUSION

This paper presents a novel In-Generation Detection (IGD) method for NSFW content detection during T2I generation. By analyzing predicted noise within diffusion models, IGD avoids reliance on text filtering or full image generation, achieving fast and accurate detection. Future work will explore finer-grained feature extraction to better distinguish semantically similar NSFW types.

## 7 ETHICS STATEMENT

This work adheres to the ICLR Code of Ethics. Our study focuses on developing a method to improve the safety of diffusion-based text-to-image models by detecting NSFW content during the generation process. The datasets used are publicly available and do not involve private or personally identifiable information. We emphasize that our proposed method is intended for enhancing AI safety and preventing harmful misuse, and we do not release any harmful prompts or unsafe image outputs. All experiments comply with ethical standards and applicable legal requirements.

## 8 REPRODUCIBILITY STATEMENT

We have made efforts to ensure the reproducibility of our results. The paper provides a detailed description of datasets, model architecture, training procedure, and evaluation metrics in Section 4 and Section 5. Additional implementation details, including hyperparameters and training setup, are provided in Section 5. We also include extended experiments and results in the Appendix to further support reproducibility.

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

# A APPENDIX

## A.1 USE OF LLMS

In preparing this paper, we made limited use of large language models (LLMs). Specifically, an LLM was employed to aid in polishing the language, improving readability, and ensuring clarity of expression. The use of LLMs was restricted to stylistic refinement and did not contribute to the conception of research ideas, methodological design, analysis, or substantive content. All contributions and intellectual work remain the responsibility of the authors.

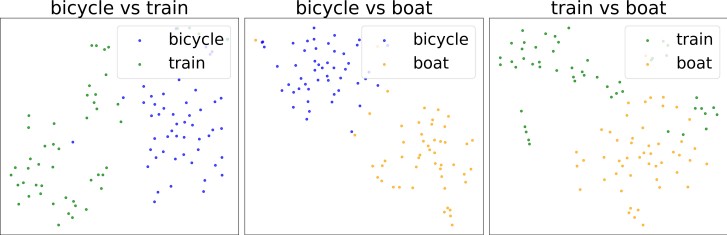

Figure 7: t-SNE visualizations of predicted noise $\epsilon_t$ for three SFW vs. SFW category pairs: bicycle vs. train, bicycle vs. boat, and train vs. boat.

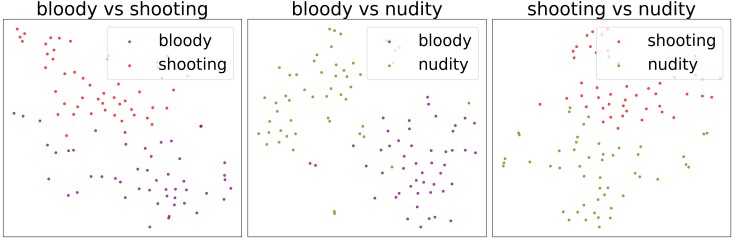

Figure 8: t-SNE visualizations of predicted noise $\epsilon_t$ for three NSFW vs. NSFW category pairs: bloody vs. shooting, bloody vs. nudity, and shooting vs. nudity.

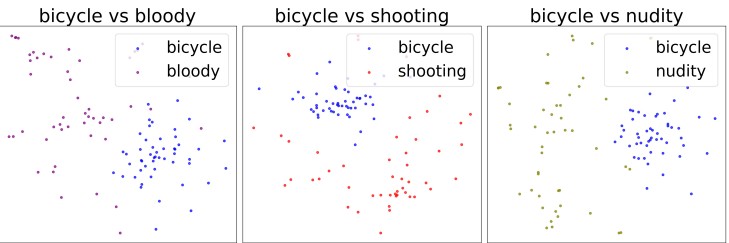

Figure 9: t-SNE visualizations of predicted noise $\epsilon_t$ for three SFW vs. NSFW category pairs: bicycle vs. bloody, bicycle vs. shooting, and bicycle vs. nudity.

## A.2 T-SNE VISUALIZATION OF PREDICTED NOISE ACROSS CATEGORIES

To examine the semantic structure encoded in the predicted noise $\epsilon_t$, we visualize its distribution across representative category pairs using t-SNE in Sec.3.2 in the main manuscript. The results are presented in Figures 7, 8, and 9.

Figure 7 illustrates intra-class comparisons among SFW categories (bicycle, boat, and train), where the predicted noise forms distinct and coherent clusters, indicating strong semantic consistency even within the SFW domain. Similarly, Figure 8 shows intra-class comparisons among NSFW categories

Table 7: Comparison with baselines on detailed types of naive NSFW prompt.

| | Methods | Naive NSFW Prompt | | | | | | | |
|---|---|---|---|---|---|---|---|---|---|
| | | sexual | violence | self-harm | harassment | hate | shocking | illegal activity | Average |
| Accuracy ↑ | NSFW-text-classifier | 57.00% | 64.50% | 64.50% | 54.50% | 62.77% | 52.00% | 65.50% | 60.11% |
| | Detoxify | 50.50% | 51.50% | 50.50% | 51.00% | 51.06% | 50.00% | 50.50% | 50.72% |
| | OpenAI Moderation API | 58.00% | 70.00% | 51.50% | 57.00% | 50.00% | 62.00% | 51.50% | 57.14% |
| | Aliyun Text Moderation | 52.00% | 50.00% | 50.00% | 50.00% | 50.00% | 50.50% | 65.50% | 52.57% |
| | Azure AI Content Safety | 63.50% | 49.50% | 57.00% | 54.50% | 51.06% | 56.50% | 57.00% | 55.58% |
| | Latent Guard | 57.00% | 60.00% | 55.00% | 56.50% | 57.45% | 55.00% | 48.50% | 55.64% |
| | GuardT2I | 54.00% | 50.00% | 55.50% | 50.50% | 50.00% | 52.00% | 50.00% | 51.71% |
| | IGD (Ours) | **89.50%** | **89.00%** | **89.50%** | **90.50%** | **95.74%** | **90.00%** | **89.00%** | **90.46%** |
| AUROC ↑ | NSFW-text-classifier | 64.82% | 65.58% | 53.46% | 65.96% | 58.40% | 58.25% | 59.42% | 60.84% |
| | Detoxify | 53.89% | 50.79% | 59.08% | 55.27% | 63.51% | 57.39% | 50.63% | 55.79% |
| | OpenAI Moderation API | 87.35% | **95.61%** | 89.92% | 83.92% | 87.46% | 92.59% | 84.43% | 88.76% |
| | Aliyun Text Moderation | 52.00% | 50.00% | 50.00% | 50.00% | 50.00% | 50.50% | 65.50% | 52.57% |
| | Azure AI Content Safety | 61.53% | 35.06% | 55.59% | 49.22% | 56.45% | 48.42% | 50.19% | 50.92% |
| | Latent Guard | 66.75% | 70.96% | 66.21% | 65.15% | 68.67% | 67.73% | 57.19% | 66.09% |
| | GuardT2I | 89.12% | 88.04% | 92.01% | 81.38% | 85.79% | 86.64% | 81.32% | 86.33% |
| | IGD (Ours) | **95.92%** | 95.35% | **95.34%** | **96.75%** | **99.68%** | **97.52%** | **93.55%** | **96.30%** |
| FPR@TPR95 ↓ | NSFW-text-classifier | 91.00% | 81.00% | 95.00% | 91.00% | 85.11% | 94.00% | 87.00% | 89.16% |
| | Detoxify | 98.00% | 100.00% | 100.00% | 100.00% | 100.00% | 95.00% | 100.00% | 99.00% |
| | OpenAI Moderation API | 44.00% | **15.00%** | 33.00% | 59.00% | 46.81% | 28.00% | 53.00% | 39.83% |
| | Aliyun Text Moderation | 100.00% | 100.00% | 100.00% | 100.00% | 100.00% | 100.00% | 100.00% | 100.00% |
| | Azure AI Content Safety | 100.00% | 100.00% | 100.00% | 100.00% | 85.11% | 100.00% | 100.00% | 97.87% |
| | Latent Guard | 76.00% | 76.00% | 78.00% | 81.00% | 89.36% | 70.00% | 81.00% | 78.77% |
| | GuardT2I | 36.00% | 45.00% | **22.00%** | 70.00% | 59.57% | 40.00% | 54.00% | 46.65% |
| | IGD (Ours) | **26.00%** | 27.00% | 29.00% | **17.00%** | **0.00%** | **11.00%** | 26.00% | **19.43%** |

Table 8: Comparison with baselines on detailed types of adversarial NSFW prompt.

| | Methods | Adversarial NSFW Prompt | | | | | | |
|---|---|---|---|---|---|---|---|---|
| | | I2P-sexual | MMA-Diffusion | Ring-A-Bell | SneakyPrompt | P4D | DiffZOO | Average |
| Accuracy ↑ | NSFW-text-classifier | 57.25% | 73.25% | 80.00% | 59.75% | 77.00% | 71.12% | 69.73% |
| | Detoxify | 50.50% | 53.00% | 67.39% | 50.25% | 57.25% | 57.75% | 56.02% |
| | OpenAI Moderation API | 57.50% | 56.50% | 93.91% | 53.75% | 70.00% | 66.98% | 66.44% |
| | Aliyun Text Moderation | 51.50% | 53.75% | 60.00% | 51.00% | 56.25% | 61.23% | 55.62% |
| | Azure AI Content Safety | 63.25% | 63.00% | 94.35% | 55.75% | 74.75% | 79.68% | 71.80% |
| | Latent Guard | 57.00% | 57.00% | 66.52% | 59.00% | 66.75% | 57.89% | 60.69% |
| | GuardT2I | 54.00% | 55.75% | **99.57%** | 52.25% | 64.00% | 67.25% | 65.47% |
| | IGD (Ours) | **94.25%** | **94.00%** | 94.78% | **93.25%** | **93.00%** | **93.32%** | **93.77%** |
| AUROC ↑ | NSFW-text-classifier | 58.92% | 63.28% | 75.39% | 54.35% | 67.22% | 63.65% | 63.80% |
| | Detoxify | 59.97% | 72.87% | 92.02% | 52.49% | 77.02% | 72.43% | 71.13% |
| | OpenAI Moderation API | 89.80% | 91.48% | **99.97%** | 85.36% | 94.37% | 94.06% | 92.51% |
| | Aliyun Text Moderation | 51.50% | 53.75% | 60.00% | 51.00% | 56.25% | 61.23% | 55.62% |
| | Azure AI Content Safety | 58.98% | 70.49% | 96.59% | 50.45% | 79.81% | 77.91% | 72.37% |
| | Latent Guard | 66.85% | 67.12% | 78.60% | 69.52% | 79.62% | 67.13% | 71.47% |
| | GuardT2I | 92.09% | 97.31% | 97.60% | 96.11% | **98.33%** | 95.58% | 96.17% |
| | IGD (Ours) | **98.00%** | **98.25%** | 99.02% | **97.81%** | 97.45% | **97.44%** | **98.00%** |
| FPR@TPR95 ↓ | NSFW-text-classifier | 94.50% | 85.50% | 71.30% | 95.00% | 82.50% | 90.11% | 86.49% |
| | Detoxify | 94.00% | 83.50% | 61.74% | 98.00% | 69.50% | 90.37% | 82.85% |
| | OpenAI Moderation API | 37.00% | 32.00% | **0.00%** | 48.00% | 30.50% | 35.83% | 30.55% |
| | Aliyun Text Moderation | 100.00% | 100.00% | 100.00% | 100.00% | 100.00% | 100.00% | 100.00% |
| | Azure AI Content Safety | 100.00% | 75.50% | 6.96% | 100.00% | 73.50% | 99.47% | 75.90% |
| | Latent Guard | 83.50% | 80.50% | 62.61% | 71.50% | 60.50% | 84.76% | 73.89% |
| | GuardT2I | 32.00% | 10.50% | 6.09% | 20.00% | **4.50%** | 18.98% | 15.35% |
| | IGD (Ours) | **6.00%** | **7.50%** | 6.09% | **7.50%** | 7.50% | **8.29%** | **7.15%** |

(bloody, shooting, and nudity), which also yield well-separated clusters, suggesting that $\epsilon_t$ reflects meaningful latent structure specific to NSFW content.

In Figure 9, the t-SNE visualizations of cross-class comparisons (SFW vs. NSFW) further reveal clear and consistent separations between the two safety categories. These results suggest that $\epsilon_t$ captures discriminative features aligned with safety semantics prior to image synthesis, highlighting its potential as an in-generation signal for NSFW detection.

## A.3 DETAILED COMPARISON WITH BASELINES ON DETAILED NSFW TYPES

As shown in Table 7, IGD achieves strong average performance on naive NSFW prompts, with 90.46% accuracy, 96.30% AUROC, and 19.43% FPR@TPR95. These results collectively demonstrate the strong generalization and fine-grained discrimination ability of IGD in detecting a broad range of inappropriate visual-textual content.

Table 9: Compare with baselines on more datasets.

|  | NSFW prompt | I2P | 4chan | Lexica | Average |
|---|---|---|---|---|---|
| **Accuracy ↑** | NSFW-text-classifier | 58.81% | 80.30% | 50.37% | 63.16% |
|  | Detoxify | 50.54% | **99.70%** | 51.86% | 67.37% |
|  | OpenAI Moderation API | 57.50% | 95.00% | 52.60% | 68.37% |
|  | Aliyun Text Moderation | 52.78% | 76.90% | 51.48% | 60.39% |
|  | Azure AI Content Safety | 56.96% | 94.70% | 53.47% | 68.37% |
|  | Latent Guard | 57.26% | 90.40% | 62.38% | 70.01% |
|  | GuardT2I | 51.70% | 74.40% | 50.87% | 58.99% |
|  | IGD (Ours) | **90.96%** | 89.70% | **93.19%** | **91.28%** |
| **AUROC ↑** | NSFW-text-classifier | 58.65% | 92.30% | 62.97% | 71.31% |
|  | Detoxify | 56.60% | **99.95%** | 48.68% | 68.41% |
|  | OpenAI Moderation API | 88.88% | 99.87% | 78.24% | 88.99% |
|  | Aliyun Text Moderation | 52.78% | 76.90% | 51.48% | 60.39% |
|  | Azure AI Content Safety | 53.44% | 96.46% | 50.50% | 66.80% |
|  | Latent Guard | 66.91% | 96.41% | 72.35% | 78.56% |
|  | GuardT2I | 87.17% | 92.71% | 87.20% | 89.03% |
|  | IGD (Ours) | **95.48%** | 94.78% | **97.06%** | **95.78%** |
| **FPR@TPR95 ↓** | NSFW-text-classifier | 90.73% | 19.20% | 91.58% | 67.17% |
|  | Detoxify | 98.30% | **0.00%** | 81.44% | 59.91% |
|  | OpenAI Moderation API | 44.51% | 0.60% | 75.00% | 40.04% |
|  | Aliyun Text Moderation | 100.00% | 100.00% | 100.00% | 100.00% |
|  | Azure AI Content Safety | 99.54% | 7.20% | 99.26% | 68.66% |
|  | Latent Guard | 85.63% | 13.40% | 77.97% | 59.00% |
|  | GuardT2I | 56.88% | 35.60% | 47.77% | 46.75% |
|  | IGD (Ours) | **26.12%** | 28.00% | **10.15%** | **21.42%** |

Table 10: Comparison on different concatenated timesteps.

| Concatenated timesteps | 5 | 5, 15, 25 | 30, 40, 50 | 5, 25, 45 | 10, 30, 50 | All |
|---|---|---|---|---|---|---|
| Accuracy ↑ | 90.96% | 89.95% | 89.49% | 89.10% | 90.19% | 97.04% |
| AUROC ↑ | 95.48% | 95.38% | 96.09% | 95.66% | 96.52% | 99.80% |
| FPR@TPR95 ↓ | 26.12% | 24.11% | 25.97% | 22.57% | 19.94% | 5.93% |

As shown in Table 8, IGD maintains high robustness under adversarial NSFW prompts, achieving 93.77% accuracy, 98.00% AUROC, and a low FPR@TPR95 of 7.15% on average. This demonstrates the strong effectiveness of our method across different adversarial NSFW prompts.

### A.4    Results on Different Datasets

To evaluate the generalization ability of our NSFW detector, we conduct experiments on two external datasets containing diverse and real-world unsafe prompts. **4chan** Qu et al. (2023) consists of 500 prompts sampled from 4chan, a web forum known for toxic discourse. These prompts are filtered based on syntactic similarity to MSCOCO captions and a high toxicity score using the Perspective API. **Lexica** Qu et al. (2023) contains 404 prompts collected from the Lexica prompt gallery, retrieved using 66 unsafe-content keywords derived from moderation guidelines and prior studies. These datasets cover a wide range of NSFW styles and serve as a robust benchmark for cross-domain evaluation.

As shown in Table 9, our method IGD achieves the best average accuracy (91.28%), AUROC (95.78%), and FPR@TPR95 (21.42%). These results demonstrate IGD's strong cross-domain detection capability, consistently outperforming baseline methods.

### A.5    Results of Classification with Information from More Timesteps

To further investigate the temporal dynamics of predicted noise $\epsilon_t$, we conduct experiments using concatenated features from multiple timesteps as input to the classifier. For example, the setting "5, 15, 25" refers to concatenating the predicted noise vectors at timesteps 5, 15, and 25 into a single representation. As shown in Table 10, several concatenated configurations achieve improved performance over using a single timestep. However, despite these gains, our method is designed with

Table 11: Results of classifiers with different numbers of MLP layers.

| MLP layers | 3 | 5 | 10 |
|---|---|---|---|
| Accuracy ↑ | 50.15% | 90.96% | 89.80% |
| AUROC ↑ | 96.47% | 95.48% | 92.84% |
| FPR@TPR95 ↓ | 16.07% | 26.12% | 45.90% |

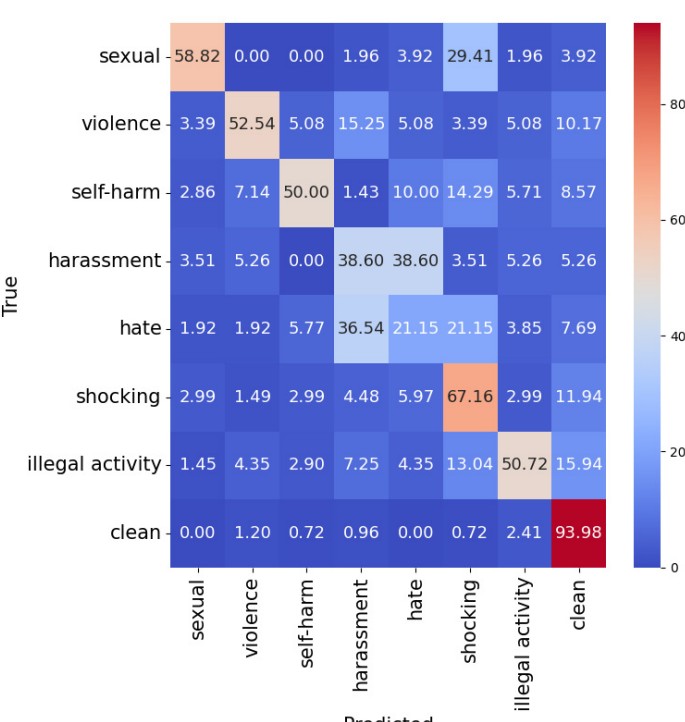

Figure 10: Confusion matrix visualization of the 8-class classification results at timestep 5.

a focus on early detection and computational efficiency. Concatenating multiple timesteps introduces additional overhead and delays prediction. Therefore, we treat these results as an experimental extension and retain the single timestep setting (t = 5) in our main pipeline. The concatenated results are presented here to support future investigations into temporal modeling strategies.

## A.6 IMPACT OF THE NUMBER OF MLP LAYERS

To assess the impact of MLP depth on classification performance, we compare models with varying numbers of layers on the naive NSFW prompt set. As shown in Table 11, moving from 3 to 5 layers yields a clear improvement in accuracy while maintaining competitive AUROC. However, further increasing the depth to 10 layers leads to degraded performance, particularly in terms of FPR@TPR95. We therefore adopt the 5-layer MLP as a balanced choice, offering strong overall performance without overfitting.

## A.7 MULTI-CLASS CLASSIFICATION ON CATEGORIES OF I2P DATASET

Beyond binary classification between NSFW and clean prompts, we further evaluate whether the model can distinguish among different types of NSFW intent. To this end, we extend the task to an 8-class classification problem, which includes seven NSFW categories from the I2P dataset along with a clean category. The confusion matrix shown in Fig. 10 illustrates the model's performance at timestep 5.

The accuracy of random-guessing on this eight-classification task is 12.5% (1/8). Our method shows a significant improvement over the random one. The clean category achieves the highest classification accuracy of 93.98%, indicating the model's strong ability to differentiate SFW from NSFW prompts. Among the NSFW classes, the shocking category achieves the highest accuracy at 67.16%, suggesting that it has the most distinct feature patterns.

These findings demonstrate that the model learns category-specific representations beyond simple binary classification, underscoring its potential for fine-grained intent recognition and adversarial content analysis.

