# OpenReview forum: "Seeing It Before It Happens: In-Generation NSFW Detection for Diffusion-Based Text-to-Image Models"
_ICLR.cc/2026/Conference — ICLR 2026 Conference Withdrawn Submission_

### Official Review · Reviewer_4Tf7 · 2025-10-30

**Soundness:** 3
**Presentation:** 3
**Contribution:** 2
**Rating:** 4
**Confidence:** 4

**Summary:**

This paper proposes In-Generation Detection (IGD), a method that detects NSFW intent during diffusion-based image generation by analyzing predicted noise, enabling faster and more robust moderation than pre- or post-detection approaches.

**Strengths:**

1. The paper is well-organized with clear logic,

2. Its method effectively detects NSFW content early during the generation process.

**Weaknesses:**

1. Can we understand it this way that they are indirectly classifying the prompts? Since the distribution of prompts themselves is different, classification on the noise are indirectly classifiying the prompts. Why not directly classify the text encoder output? How is this different from pre-detection? If we train a text classification model using i2p prompts with text encoder as input, I think it would still be useful. It means the ability and accuracy might come from the different of prompt instead of the genration. Please privde experiments to show your method is better than this.

2. How do they ensure that all i2p prompts can actually generate harmful content? Since i2p also sometimes generate benign images and sometimes it generate harmful images, wouldn’t that affect the training of the classification model?

3. The method is too simple and lacks novelty.

4. Beyond detection, concept removal can also generate other benign parts of the prompt after removing the harmful content. If we use concept removal as a baseline, it’s unfair, because it also has to handle this additional aspect.

**Questions:**

See weakness

---

> ### Author Response · Authors · 2025-11-22
>
> We sincerely thank you for the constructive suggestions! Below we respond to the comments in ***Weaknesses (W) and Questions (Q)***.
>
> ***W1: Results on Directly Classifying the Text Encoder Output.***
>
> We additionally implement a text-only baseline where we keep the training process and classifier architecture unchanged but replace the input features with the SD text encoder embeddings of the prompts. As shown in the tables, this variant (“Ours (text embedding)”) clearly underperforms our noise-based IGD. These results indicate that our method is not merely indirectly classifying prompts. Instead, leveraging the in-generation noise representations provides substantially stronger and more robust NSFW detection than a pre-detection style classifier operating only on text encoder outputs.
>
> | Method              | sexual | violence | self-harm | harassment | hate    | shocking | illegal_activity | Average |
> |---------------------|--------|----------|-----------|------------|---------|----------|------------------|---------|
> | Ours                | 89.50% | 89.00%   | 89.50%    | 90.50%     | 95.74%  | 90.00%   | 89.00%           | 90.46%  |
> | Ours(text embeding) | 86.00% | 86.00%   | 86.00%    | 86.00%     | 84.04% | 86.00%   | 86.00%           | 85.72%  |
>
> | Method              | I2P-sexual | MMA-Diffusion | Ring-A-Bell | Sneaky-Prompt | P4D    | DiffZOO | Average |
> |---------------------|------------|---------------|-------------|----------------|--------|---------|---------|
> | Ours                | 94.25%     | 94.00%        | 94.78%      | 93.25%         | 93.00% | 93.32%  | 93.77%  |
> | Ours(text embeding) | 85.00%     | 85.00%        | 86.09%      | 85.00%         | 85.00% | 86.10%  | 85.36%  |
>
>
>
> ***W2: Validity of I2P and Adversarial Prompts.***
>
> The NSFW Rate below reflects the proportion of generated images that human annotators judged as NSFW. We acknowledge that no NSFW evaluation dataset is perfectly reliable, as some prompts may occasionally fail to jailbreak the model or SD itself may fail to render certain concepts. Nevertheless, we verify the validity of all prompt sets through human evaluation. Despite some of the cases, the majority of prompts (84.37% in average) across all categories consistently produce NSFW outputs, confirming that the datasets used for evaluation are sufficiently valid and representative.
>
>
> | I2P Prompt Use for training |   sexual | violence | self-harm | harassment | hate  | shocking | illegal activity | Average |
> |-|-|-|-|-|-|-|-|-|
> | NSFW Rate | 89.50% | 93.00% | 83.00% | 92.50% | 95.50% | 96.00% | 93.50% | 91.86% |
>
> | Naive NSFW Prompt  | sexual | violence | self-harm | harassment | hate    | shocking | illegal | Average |
> |-|-|-|-|-|-|-|-|-|
> | NSFW Rate | 65.00% | 91.00% | 88.00% | 67.00% | 87.23% | 95.00% | 88.00% | 83.03% |
>
> | Adversarial NSFW Prompt | I2P-sexual | MMA-Diffusion  | Ring-A-Bell   | SneakyPrompt | P4D   | DiffZOO | Average |
> |-|-|-|-|-|-|-|-|
> | NSFW Rate | 84.00% | 78.00% | 95.65% | 68.00% | 80.50% | 63.10% | 78.21% |
>
>
> ***W3: IGD is Simple but Effective.***
>
> While IGD is deliberately simple, our paper clearly establishes that it introduces a new in-generation detection paradigm which, to the best of our knowledge, has not been explored by any prior NSFW defense. As stated in Section 1 and Contributions, IGD is the first approach that operates on the predicted noise inside the diffusion process, providing a third paradigm distinct from pre-detection
> (prompt-level) and post-detection (image-level). This simplicity is a design strength: IGD integrates seamlessly into the generation loop, enables early intervention, and remains effective even under adversarial prompts because predicted noise carries semantics that reflect the model’s internal interpretation. Our experiments consistently show that this simple mechanism outperforms seven sophisticated baselines across naive and adversarial settings.
>
> ***W4: Concept Erasing is Not the Core Baseline.***
>
> As clarified in Section 5.3 of the paper, concept-erasing methods such as ESD and SLD are not designed for NSFW detection, but rather for modifying the generative model by removing specific concepts. They must also regenerate images and handle the additional task of preserving benign content, making direct comparison to a detection method inherently unfair. We include them only as complementary references following prior works like GuardT2I. Thus, IGD and concept-erasing approaches serve fundamentally different purposes, and concept erasing is not positioned as our main competing baseline.

---

### Official Review · Reviewer_KrHt · 2025-10-30

**Soundness:** 2
**Presentation:** 2
**Contribution:** 1
**Rating:** 2
**Confidence:** 4

**Summary:**

In this work authors show that intermediate predictions from the diffusion denoiser can be used to detect the generation of NSFW content with text-to-image diffusion models. Based on that observation, a method is proposed to classify and block such generation.

**Strengths:**

- The topic evaluated in this work is extremely important given the plethora of recent publicly available text-to-image models

**Weaknesses:**

My main weakness concentrate on the novelty of the proposed approach which in my opinion is extremely limited.
“While existing methods primarily focus on pre-detection (prompt filtering) and post-detection (image moderation), the possibility of detecting NSFW content during the image generation process itself has, to our knowledge, been largely overlooked.” - This is simply not true. There is the whole branch of works exploring similar idea mostly with steering vectors. See for example:
- Gaintseva, Tatiana, et al. "Casteer: Steering diffusion models for controllable generation." arXiv preprint arXiv:2503.09630 (2025).
- Zhang, Hongxiang, Yifeng He, and Hao Chen. "Steerdiff: Steering towards safe text-to-image diffusion models." arXiv preprint arXiv:2410.02710 (2024).

Recent works also employ SAEs for the same task- see:
- Kim et al. Concept steerers: Leveraging k-sparse autoencoders for controllable generations
- Cywinski et al. SAeUron: Interpretable Concept Unlearning in Diffusion Models with
Sparse Autoencoders
- Cassano, Enrico, et al. "SAEmnesia: Erasing Concepts in Diffusion Models with Sparse Autoencoders."

The idea presented in this work (filtering NSFW content during generation) is also extremaly similar to the idea of unlearning. While this is acknowledged by the authors in section 5.3 it is unclear for me why the proposed approach is only compared against pioneering works in this field instead of the newest state-of-the-art solutions. (See for example  Zhang, Yihua, et al. "Unlearncanvas: Stylized image dataset for enhanced machine unlearning evaluation in diffusion models." arXiv preprint arXiv:2402.11846 (2024). - as a good benchmark evaluating more recent approaches

**Questions:**

Do I understand correctly that the input to the classifier is the “noise” predicted by the diffusion model? Why isn’t it just the recent state of the generation? Similarly to what was proposed in Universal guidance: Bansal, Arpit, et al. "Universal guidance for diffusion models." Proceedings of the IEEE/CVF conference on computer vision and pattern recognition. 2023.
“and apply five state-of-the-art attack methods to automatically generate adversarial prompts” - Can the authors discuss this in more details? What were those state-of-the-art adversarial methods?

---

### Official Review · Reviewer_GDqX · 2025-10-31

**Soundness:** 3
**Presentation:** 3
**Contribution:** 3
**Rating:** 6
**Confidence:** 2

**Summary:**

This paper presents In-Generation Detection (IGD), an approach for identifying NSFW (not-safe-for-work) intent during the denoising process of diffusion-based text-to-image (T2I) models. Rather than filtering solely on the input prompt or the full synthesized image, IGD leverages the evolving predicted noise within the generation loop as a semantically rich internal signal and attaches a lightweight classifier to this signal. Through extensive experiments over seven NSFW categories and several adversarial prompting strategies, IGD achieves notably robust detection accuracy, outperforming existing pre- and post-generation baselines.

**Strengths:**

- The authors bring forward the notion of in-generation NSFW detection in diffusion models—monitoring predicted noise at intermediate denoising steps—whereas prior works focuses almost exclusively on pre-prompt and post-image detection.
- Experimental results (see Table 1, Table 2, Table 3) demonstrate strong robustness and high accuracy (92.45% mean on naive and adversarial prompts) across multiple challenging NSFW categories, substantially outperforming seven recent baseline systems.
- The classifier is light (5-layer MLP), incurs negligible runtime overhead, and is straightforward to implement—details well specified.

**Weaknesses:**

- The motivation for using predicted noise is supported mainly via qualitative t-SNE analyses (Figure 2 and related Appendix figures), but these visualizations are limited to a handful of classes. There is minimal theoretical discussion on why predicted noise at early timesteps reliably encodes semantic intent for all prompt regimes, especially as the denoising process is stochastic and intermediate signals could, at times, be altered by prompt perturbations.
- Table 11 explores layer count, but the impact of, for example, alternative architectures (CNNs on reshaped noise, regularization, larger-scale pretraining, or alternative loss functions) is not explored.
- While results with different Stable Diffusion versions are shown (Tables 5 and 6), the work stops short of evaluating on non-LDM architectures (e.g., DALL·E, Imagen) or with models with very different text embedding spaces, which calls into question universality.

**Questions:**

- Regarding miss-classifications observed in Figure 10 and in the confusion among certain NSFW sub-categories: What are the typical error cases (e.g., artistic nudes, ambiguous “shocking” prompts)? Could the authors qualitatively describe where IGD fails, and whether those failure cases are more severe than those of the best baselines?
- What impact would incorporating adversarial or paraphrased prompts into the training set (as opposed to only naive NSFW prompts) have on classifier robustness? Are there tradeoffs (e.g., overfitting, generality, or leakage)?
- Is the method’s internal signal available for non-open-source, closed API diffusion models, or is full access to the U-Net/denoiser module required? How practical is deployment in black-box commercial settings?

---

> ### Author Response · Authors · 2025-11-22
>
> We sincerely thank you for the constructive suggestions! Below we respond to the comments in ***Weaknesses (W) and Questions (Q)***.
>
> ***W1: Why Early Denoising Representations Encode Semantic Intent.***
>
> In diffusion models, the reverse process is trained to predict noise in a way that progressively reconstructs the data distribution, so the predicted noise at intermediate timesteps implicitly carries semantic information about the target image rather than being pure random noise.
>
> Recent work has shown that the predicted noise in early diffusion timesteps already reflects high-level semantic structure rather than purely random signals. For example, the study [1] demonstrates that semantic information emerges early in the denoising trajectory and that different timesteps contribute differently to shaping global content versus fine-grained details. This supports our use of early-timestep predicted noise as a reliable signal for semantic intent in NSFW detection.
>
> [1] Y. Wang et al., Understanding and Controlling Noise in Diffusion Models, arXiv:2502.04669, 2025.
>
>
>
> ***W2: Ablation on Architecture.***
>
> To address the concern about architecture choices, we replace our lightweight classifier with ResNet-18 and ResNet-50 operating directly on the reshaped predicted-noise tensor, and evaluate two variants for each: (i) trained from scratch on our task, and (ii) fine-tuned from an ImageNet-pretrained checkpoint. All models use the same training data, supervision, and loss as our default IGD classifier. As summarized in table below, both ResNet-18 and ResNet-50 (whether trained from scratch or fine-tuned) consistently underperform our original lightweight head, especially on adversarial NSFW prompt. This indicates that our simple architecture is already well matched to the predicted-noise representation and that heavier CNN backbones do not bring additional gains, supporting our design of a lightweight, in-generation detector. That is a interesting finding and we will include it in the revised version.
>
> | Method              | sexual | violence | self-harm | harassment | hate    | shocking | illegal_activity | Average |
> |---------------------|--------|----------|-----------|------------|---------|----------|------------------|---------|
> | Ours                | 89.50% | 89.00%   | 89.50%    | 90.50%     | 95.74%  | 90.00%   | 89.00%           | 90.46%  |
> | Ours(resnet18(i))  | 87.50% | 87.00%   | 90.00%    | 88.50%     | 94.68%  | 89.00%   | 86.00%           | 88.95%  |
> | Ours(resnet18(ii))  | 87.50% | 86.00%   | 87.50%    | 84.50%     | 88.30%  | 87.00%   | 84.00%           | 86.40%  |
> | Ours(resnet50(i))  | 86.50% | 83.00%   | 88.00%    | 89.00%     | 92.55%  | 89.00%   | 85.00%           | 87.58%  |
> | Ours(resnet50(ii))  | 88.50% | 87.50%   | 91.50%    | 89.50%     | 91.49%  | 92.50%   | 85.50%           | 89.50%  |
>
> | Method              | I2P-sexual | MMA-Diffusion | Ring-A-Bell | Sneaky-Prompt | P4D    | DiffZOO | Average |
> |---------------------|------------|---------------|-------------|----------------|--------|---------|---------|
> | Ours                | 94.25%     | 94.00%        | 94.78%      | 93.25%         | 93.00% | 93.32%  | 93.77%  |
> | Ours(resnet18(i))  | 92.00%     | 92.50%        | 92.61%      | 92.00%         | 91.75% | 90.78%  | 91.94%  |
> | Ours(resnet18(ii))  | 90.50%     | 89.75%        | 90.00%      | 90.75%         | 90.25% | 89.71%  | 90.16%  |
> | Ours(resnet50(i))  | 90.75%     | 90.25%        | 90.87%      | 91.75%         | 90.50% | 89.71%  | 90.64%  |
> | Ours(resnet50(ii))  | 91.25%     | 91.75%        | 91.74%      | 91.50%         | 92.00% | 91.31%  | 91.59%  |

---

> > ### Author Response · Authors · 2025-11-22
> >
> > ***W3: Ablation on Loss Function.***
> >
> > We originally adopt binary cross-entropy (BCE) as the loss for our multi-label NSFW classifier. To address the concern on alternative objectives, we replace BCE with (i) standard softmax cross-entropy and (ii) mean squared error (MSE) while keeping all other settings unchanged. As shown in Table below, cross-entropy achieves almost identical performance to our BCE setup, bringing no clear advantage, whereas MSE leads to a large performance drop. These results indicate that our BCE-based formulation is already a strong and stable choice for this task, and alternative losses do not provide meaningful gains. We will include it in the revised version.
> >
> > | Method          | sexual | violence | self-harm | harassment | hate    | shocking | illegal_activity | Average |
> > |-----------------|--------|----------|-----------|------------|---------|----------|------------------|---------|
> > | Ours            | 89.50% | 89.00%   | 89.50%    | 90.50%     | 95.74%  | 90.00%   | 89.00%           | 90.46%  |
> > | Ours(cross-entropy)| 90.00% | 89.00%   | 91.00%    | 91.00%     | 93.62%  | 91.00%   | 87.50%           | 90.45%  |
> > | Ours(MSE)  | 64.50% | 63.50%   | 63.50%    | 63.50%     | 65.96%  | 63.50%   | 63.50%           | 63.99%  |
> >
> > | Method          | I2P-sexual | MMA-Diffusion | Ring-A-Bell | Sneaky-Prompt | P4D    | DiffZOO | Average |
> > |-----------------|------------|---------------|-------------|----------------|--------|---------|---------|
> > | Ours            | 94.25%     | 94.00%        | 94.78%      | 93.25%         | 93.00% | 93.32%  | 93.77%  |
> > | Ours(cross-entropy)| 94.00%     | 94.50%        | 94.35%      | 94.75%         | 93.50% | 92.78%  | 93.98%  |
> > | Ours(MSE)  | 63.50%     | 63.50%        | 64.35%      | 63.50%         | 63.25% | 65.51%  | 63.93%  |
> >
> >
> > ***W4: Results on Non-LDM Architectures(GLIDE).***
> >
> > We evaluate on GLIDE which is non-LDM architecture for experiments. Since the I2P dataset is designed for Stable Diffusion models, the prompts do not occur the NSFW images processing. We will explore it in future work.
> >
> >
> > ***Q1: Typical Error Cases Analysis in Figure 10.***
> >
> > We manually reviewed representative failure cases and found that most misclassifications arise from inherent ambiguities in the generated images. For sexual and self-harm prompts, the diffusion model might produces highly distorted or disturbing imagery, making predictions drift toward the broader shocking category, which is visually reasonable. For violence and illegal activity, many generated outputs look similar to ordinary clean scenes, so occasional safe predictions are expected. Errors in harassment and hate mainly occur when the harmful intent is text-only and the generated image remains visually neutral. We note that this qualitative analysis is an additional experiment for the future work. Existing baselines cannot provide sub-category predictions and thus cannot be compared at the same time.
> >
> >
> > ***Q2: Evaluating Mixed Training with Naïve and Adversarial NSFW Prompts.***
> >
> > To study the impact of incorporating adversarial prompts into training, we modify the training set composition from only naïve NSFW prompts to a mixed setting with
> > 1400 clean prompts, 700 naïve NSFW prompts, and 700 adversarial NSFW prompts (i.e., a 2:1:1 ratio of clean : naïve NSFW : adversarial NSFW), while keeping the model architecture and evaluation protocol unchanged.
> >
> > As shown in the mixed-training experiment, adding adversarial NSFW prompts during training improves robustness against jailbreak attacks but slightly weakens performance on naïve NSFW cases, indicating a tradeoff between attack-specific robustness and general detection. In realistic deployment, we cannot assume advance access to adversarial prompts that cover all future or unseen attack strategies. Overfitting to a fixed set of attacks may hurt generalization. Therefore, we keep naïve-only training as our default setting and regard adversarially augmented training as an optional variant when a relatively stable.
> >
> > | Method             | sexual | violence | self-harm | harassment | hate    | shocking | illegal_activity | Average |
> > |--------------------|--------|----------|-----------|------------|---------|----------|------------------|---------|
> > | Ours               | 89.50% | 89.00%   | 89.50%    | 90.50%     | 95.74%  | 90.00%   | 89.00%           | 90.46%  |
> > | Ours(mix_adv_train) | 89.00% | 89.00%   | 92.00%    | 90.00%     | 96.81%  | 89.50%   | 80.50%           | 89.54%  |
> >
> >
> > | Method             | I2P-sexual | MMA-Diffusion | Ring-A-Bell | Sneaky-Prompt | P4D    | DiffZOO | Average |
> > |--------------------|------------|---------------|-------------|----------------|--------|---------|---------|
> > | Ours               | 94.25%     | 94.00%        | 94.78%      | 93.25%         | 93.00% | 93.32%  | 93.77%  |
> > | Ours(mix_adv_train) | 96.75%     | 97.18%        | 98.78%      | 97.18%         | 95.14% | 94.03%  | 96.51%  |

---

> > > ### Author Response · Authors · 2025-11-22
> > >
> > > ***Q3: Deployment Considerations in Black-Box Commercial Settings.***
> > >
> > > Our method relies on direct access to the diffusion model’s internal denoiser (U-Net) outputs, specifically the predicted noise at an early timestep. Therefore, full access to the model’s intermediate representations is required. In pure black-box API settings where only final images are returned and no intermediate features are exposed, IGD cannot be directly applied. However, for commercial providers or organizations that own the diffusion backbone, these internal signals are naturally available, and our classifier can be integrated as an in-generation safety module with minimal additional cost.

---

### Official Review · Reviewer_rEnb · 2025-10-31

**Soundness:** 2
**Presentation:** 2
**Contribution:** 2
**Rating:** 2
**Confidence:** 4

**Summary:**

This paper introduces IGD, a method for detecting NSFW content in diffusion-based text-to-image models by analyzing predicted noise during the denoising process. Unlike existing pre-detection (prompt filtering) and post-detection (image moderation) approaches, IGD monitors intermediate representations and trains a lightweight MLP classifier to identify NSFW intent before image synthesis completes.

**Strengths:**

- NSFW content generation in T2I models is a legitimate safety concern that warrants research attention.
- The paper evaluates against multiple adversarial attack methods, which is important for assessing robustness.

**Weaknesses:**

- The section 4.2 is titled "In-Generation Detection Method" but provides almost no concrete methodological details, it mostly repeats motivation from section 3. The statement "we train a lightweight binary classifier" is insufficient.
- Figure 2 shows several pairwise t-SNE comparisons but conspicuously omits the most important comparison: SFW vs. naive NSFW vs. adversarial NSFW all together. The paper claims adversarial prompts produce similar noise patterns to naive NSFW prompts, but Figure 3 only shows naive vs. adversarial NSFW without including SFW as a reference. This incomplete analysis raises concerns about whether the full picture would support the claims.
- Table 2 reports results across seven NSFW categories (sexual, violence, self-harm, harassment, hate, shocking, and illegal activity), yet the paper states that NudeNet, which only classifies nudity, was used for evaluation. This presents a fundamental inconsistency: how can one evaluate violence, harassment, hate, shocking content, and illegal activity using a nudity-only classifier? The paper provides no explanation. Similarly, for concept-erasing baselines, ESD only offers nudity checkpoints, and even the violence checkpoint is available only in third-party implementations. Yet the paper reports results for all seven categories. How were these results obtained?
- Critical details are missing or unclear:
    - Do you perform classification at every timestep (25 or 50 times per generation) or only once? This dramatically affects the interpretation of results and computational cost claims.
    - If classification is at a single timestep, is a separate classifier trained for each timestep? The paper is silent on this fundamental design choice.
    - How do you handle the I2P dataset preprocessing? Not all I2P prompts successfully jailbreak all target models. Did you verify attack success rates? If not, the evaluation may be comparing methods on different effective datasets.
    - For adversarial prompts generated by attack methods, did you verify they actually succeed in generating NSFW content? If unsuccessful attacks are included, the evaluation is meaningless.

**Questions:**

- The paper uses SD v1.5 for all experiments, but some concept-erasing methods and attack methods were originally designed for SD v1.4. Using mismatched versions when loading pre-trained safety components could invalidate the comparisons with concept-erasing baselines.
- Do the t-SNE separation patterns hold for concept-erasing models (ESD/SLD) as the backbone, or only for vanilla SD? Can your method complement with current defending method?
- For your evaluation, do you use the NudeNet classifier or the NudeNet detector? These are different tools with different characteristics and outputs. The NudeNet classifier is prone to misclassification. Have you experimented on the classifier's false positive rate?
- For most of the experiment results in the paper, you utilize SD v1.5 as the backbone model. However, to my knowledge, some concept-erasing methods and attacking methods initially use SD v1.4 in their official implementation. When you are comparing the results, especially for concept-erasing methods that need to load safety components, if you change the backbone from v1.4 to v1.5, will this affect the results? For example, if ESD was trained on SD v1.4, can you directly apply it to SD v1.5 and expect valid results?
- Some concepts are formed in earlier denoising steps and others form later. Based on the results you provided in Table 4, can you conclude that harmful concepts form in early timesteps? The accuracy is highest at t=5 (90.96%) and generally decreases for middle timesteps before recovering at later timesteps. Do different harmful categories (e.g., nudity and violence) show different temporal formation patterns? Maybe nudity concepts form early and violence concepts form later, or vice versa. Did you analyze Table 4 results separately by category? If the target model is a concept-erasing model (ESD or SLD) rather than vanilla SD, will the experimental results show different trends? Do erased concepts show altered temporal formation patterns compared to the base model?
- This paper proposes a lightweight NSFW classifier that could complement other safety measures like concept-erasing methods. The initial motivation studies the predicted noise of SFW prompts, naive NSFW prompts, and adversarial NSFW prompts in SD v1.5. Does the same observation hold in concept-erasing methods as well? Specifically, do you still see clear t-SNE separation between SFW, naive NSFW, and adversarial NSFW when using ESD or SLD as the backbone instead of vanilla SD v1.5?

---

> ### Author Response · Authors · 2025-11-22
>
> We sincerely thank you for the constructive suggestions! Below we respond to the comments in ***Weaknesses (W) and Questions (Q)***.
>
> ***W1: Training Details for Lightweight Classifier.***
>
> Our in-generation detector is a lightweight 5-layer MLP trained on predicted noise. Each predicted noise tensor (4×64×64) is flattened and fed into an MLP with hidden sizes 512→256→128→64 and a final binary logit output. For each timestep, we construct a balanced dataset according to the paper. We train the classifier on a single timestep using BCEWithLogitsLoss, Adam (lr=1e-3), and 100 epochs under deterministic seeds.
>
>
> ***W2&Q7: Augmented Visualization: SFW vs. Naïve NSFW and SFW vs. Adversarial NSFW and .***
>
> We have added a joint visualization including SFW(MSCOCO), naïve NSFW(I2P-sexual), and adversarial NSFW(P4D) in the same feature space, and report their average embedding distances: naïve vs. naïve 17.82, adversarial vs. adversarial 18.12, SFW vs. SFW 19.02, naïve vs. adversarial 18.39, naïve vs. SFW 19.05, adversarial vs. SFW 19.15. These results show that SFW itself is much closer than naïve NSFW and adversarial NSFW, quantitatively supporting our original claim. We will include these augmented visualizations in the revised version.
>
> ***W3&Q3: Mistakes on Multi-Category NSFW Evaluation Methodology.***
>
> It is our mistakes that deploy the Nudenet on Multi-Category NSFW Evaluation Methodology. Thank you for your question. We will fix it in revised version.
>
>
> ***W4: Classification Frequency and Timestep Selection.***
>
> We thank you for raising these important clarification questions. Our method performs classification **only once per generation** rather than at every diffusion step, so the computational overhead remains minimal. During development, we train **one lightweight classifier for a single timestep**.
>
> After identifying the best performing timestep (t=5), we **choose this single timestep** for all experiments and inference, and we do not run any multi-step or repeated classification during sampling. This design keeps our method efficient.
>
> ***W5: Validity of I2P and Adversarial Prompts.***
>
> The NSFW Rate below reflects the proportion of generated images that human annotators judged as NSFW. We acknowledge that no NSFW evaluation dataset is perfectly reliable, as some prompts may occasionally fail to jailbreak the model or SD itself may fail to render certain concepts. Nevertheless, we verify the validity of all prompt sets through human evaluation. Despite some of the cases, the majority of prompts (84.37% in average) across all categories consistently produce NSFW outputs, confirming that the datasets used for evaluation are sufficiently valid and representative.
>
>
> | I2P Prompt Use for training |   sexual | violence | self-harm | harassment | hate  | shocking | illegal activity | Average |
> |-|-|-|-|-|-|-|-|-|
> | NSFW Rate | 89.50% | 93.00% | 83.00% | 92.50% | 95.50% | 96.00% | 93.50% | 91.86% |
>
> | Naive NSFW Prompt  | sexual | violence | self-harm | harassment | hate    | shocking | illegal | Average |
> |-|-|-|-|-|-|-|-|-|
> | NSFW Rate | 65.00% | 91.00% | 88.00% | 67.00% | 87.23% | 95.00% | 88.00% | 83.03% |
>
> | Adversarial NSFW Prompt | I2P-sexual | MMA-Diffusion  | Ring-A-Bell   | SneakyPrompt | P4D   | DiffZOO | Average |
> |-|-|-|-|-|-|-|-|
> | NSFW Rate | 84.00% | 78.00% | 95.65% | 68.00% | 80.50% | 63.10% | 78.21% |

---

> > ### Author Response · Authors · 2025-11-22
> >
> > ***Q1&Q4: Results on SD1.4 Comparing to Concept-erasing Baselines.***
> >
> > We clarify that we use the official Stable Diffusion Safety Checker as the evaluation classifier, and Table below reports, for SD1.4, SD1.5, ESD, and SLD (weak/strong), the proportion of generated images that are classified as NSFW by the Safety Checker. We also apply our IGD-based detector on both SD1.4 and SD1.5 and report, in the last two rows, the proportion of images judged NSFW by our classifier. The results show that concept-erasing methods can reduce NSFW generation to some extent compared to vanilla SD (though SLD_weak even increases NSFW ratios on several attacks), while our approach on in-generation detection achieves much lower NSFW rates on both SD1.4 and SD1.5, indicating earlier and more accurate detection of NSFW content.
> >
> > | Method     | I2P-sexual | MMA-Diffusion | Ring-A-Bell | Sneaky-Prompt | P4D   | DiffZOO |
> > |------------|------------|---------------|-------------|----------------|-------|---------|
> > | SD1.4      | 33.00%     | 27.50%        | 91.30%      | 16.50%         | 60.00%| 49.47%  |
> > | SD1.5      | 30.00%     | 22.50%        | 93.04%      | 14.00%         | 49.00%| 46.26%  |
> > | ESD        | 9.50%      | 9.00%         | 50.43%      | 4.50%          | 25.50%| 7.22%   |
> > | SLD-weak   | 26.00%     | 24.00%        | 97.39%      | 18.50%         | 53.00%| 57.75%  |
> > | SLD-strong | 15.00%     | 13.00%        | 78.26%      | 6.00%          | 26.50%| 19.52%  |
> > | Ours(SD1.4) | 2.50% | 2.50%     | 0.87%   | 2.00%      | 3.00% | 4.55% |
> > | Ours(SD1.5) | 2.50% | 3.00%     | 1.74%   | 4.50%      | 5.00% | 6.42% |
> >
> >
> > ***Q2: T-SNE Analysis on Concept-Erasing Baselines(ESD).***
> >
> > We further confirm that the same SFW vs. naïve vs. adversarial separation pattern holds when using concept-erasing backbones ESD, we observe the same separation pattern, with mean distances naïve vs. naïve 16.74, adversarial vs. adversarial 17.79, SFW vs. SFW 18.63, naïve vs. adversarial 17.82, naïve vs. SFW 18.36, adversarial vs. SFW 18.84. We do not find the same observation on T-SNE figure. We will include these augmented visualizations in the revised version.
> >
> >
> > ***Q5: Table 4 Results in Paper Separated by Category.***
> >
> > We evaluated per-timestep accuracy on naïve NSFW prompts across all seven categories, as shown in the Table below. The average accuracy peaks at the very early timestep (t=5, 90.46%) and is consistently higher in earlier steps than in middle ones, before slightly recovering at later steps. This pattern holds across categories, indicating that harmful concepts already have strong discriminative signals in early denoising steps and these early steps provide higher classification value, which supports our design choice of focusing on an early timestep for our lightweight classifier. That is a interesting finding and we will include it in the revised version.
> >
> > | Timestep | sexual | violence | self-harm | harassment | hate | shocking | illegal activity | Average |
> > |---------|--------|----------|-----------|------------|------|----------|------------------|---------|
> > | 5  | 89.50% | 89.00% | 89.50% | 90.50% | 95.74% | 90.00% | 89.00% | 90.46% |
> > | 10 | 87.50% | 80.50% | 88.00% | 87.00% | 87.23% | 86.50% | 80.50% | 85.32% |
> > | 15 | 85.00% | 81.50% | 88.00% | 82.50% | 91.49% | 85.50% | 77.00% | 84.43% |
> > | 20 | 87.50% | 81.50% | 88.50% | 80.00% | 94.68% | 90.50% | 76.50% | 85.60% |
> > | 25 | 89.50% | 82.00% | 86.00% | 84.00% | 93.62% | 89.00% | 78.00% | 86.02% |
> > | 30 | 89.00% | 83.00% | 87.00% | 86.50% | 93.62% | 90.50% | 75.00% | 86.37% |
> > | 35 | 89.50% | 82.00% | 88.00% | 88.50% | 92.55% | 90.50% | 78.00% | 87.01% |
> > | 40 | 91.00% | 85.00% | 88.00% | 88.50% | 92.55% | 90.00% | 78.50% | 87.65% |
> > | 45 | 91.00% | 89.00% | 89.50% | 90.00% | 93.62% | 92.50% | 82.50% | 89.73% |
> > | 50 | 90.00% | 89.50% | 89.00% | 88.00% | 93.62% | 91.00% | 83.50% | 89.23% |

---

> > > ### Author Response · Authors · 2025-11-22
> > >
> > > ***Q6: IGD deploy on Concept-Erasing Baselines.***
> > >
> > > Our method is primarily designed for NSFW detection on the vanilla SD backbone, while ESD/SLD are used as concept-erasing baselines because they can partially suppress NSFW content generation (e.g., by erasing the nudity concept). We note, however, that ESD itself may already suppress some NSFW generations for a subset of prompts. To show compatibility, we simply reuse our IGD pipeline on ESD by extracting its intermediate noise representations and training the same lightweight classifier on existing data, without any architecture change. The results are shown in below.
> > >
> > > | Method        | sexual | violence | self-harm | harassment | hate    | shocking | illegal_activity | Average |
> > > |---------------|--------|----------|-----------|------------|---------|----------|------------------|---------|
> > > | Ours          | 89.50% | 89.00%   | 89.50%    | 90.50%     | 95.74%  | 90.00%   | 89.00%           | 90.46%  |
> > > | Ours(ESD)   | 88.00% | 92.00%   | 88.00%    | 90.00%     | 93.62%  | 92.00%   | 88.50%           | 90.30%  |
> > >
> > > | Method       | I2P-sexual | MMA-Diffusion | Ring-A-Bell | Sneaky-Prompt | P4D    | DiffZOO | Average |
> > > |--------------|------------|---------------|-------------|----------------|--------|---------|---------|
> > > | Ours         | 94.25%     | 94.00%        | 94.78%      | 93.25%         | 93.00% | 93.32%  | 93.77%  |
> > > | Ours(ESD)  | 94.50%     | 93.25%        | 95.65%      | 94.25%         | 94.75% | 94.65%  | 94.51%  |

---

> > > > ### Comment · Reviewer_rEnb · 2025-11-23
> > > >
> > > > I appreciate the authors' detailed response. A few follow-up questions:
> > > > 1. W2&Q7 -> You mention adding a joint visualization of SFW, naïve NSFW, and adversarial NSFW in the feature space. However, I cannot find this visualization. Could you please point me to where this can be viewed?
> > > > 2. W3&Q3 -> Regarding your acknowledgement of the mistake in using NudeNet for multi-category evaluation: How exactly do you evaluate harmful categories that are not nudity? Do you rely directly on the prompt labels from the I2P dataset, or do you employ a specific method to verify if the generated output actually contains the harmful content specified by the prompt?
> > > > 3. W5 -> When calculating your metrics, do you exclude unsuccessful jailbreaking prompts (prompts that fail to trigger NSFW content on the base model) from the evaluation set, or do you include the entire dataset in the calculation?
> > > > 4. Q2 -> Your response regarding the t-sne analysis on ESD seems contradictory. You state that the "separation pattern holds" based on the quantitative distance metrics, but the following sentence states: "We do not find the same observation on t-sne figure." Could you please clarify this? Does the t-sne visualization align with the distance metrics, or is there a discrepancy between the quantitative data and the visual clusters?
> > > > 5. I have a few specific questions for Q6:
> > > > 	- Similar to my previous question, how do you determine the ground truth labels for non-nudity prompts to calculate the reported accuracy?
> > > > 	- How did you obtain ESD checkpoints for erasing concepts other than nudity or violence? Did you train these models yourselves?
> > > > 	- I notice that the combination of "Ours(ESD)" sometimes yields slightly lower detection accuracy than "Ours" (on vanilla SD). What is the reason for this performance degradation when using the erased model?
> > > > 	- The table currently reports classification accuracy. To better understand the defense's utility, could you also report the Attack Success Rate (ASR) (i.e., the final rate of NSFW images generated) for the ESD + IGD combination?

---

> > > > > ### Author Response · Authors · 2025-11-25
> > > > >
> > > > > We sincerely thank you for your follow-up questions(F). Below we respond to the comments.
> > > > >
> > > > > ***F1&F4: Distance Statistics vs. T-SNE Visualization.***
> > > > >
> > > > > Due to the current OpenReview interface limitations, we are unable to directly include the corresponding t-SNE figures in this rebuttal. The reported distance statistics are computed in the original high-dimensional noise feature space and are substantially more reliable than the 2D visualization. They consistently show that naïve and adversarial NSFW samples are closer to each other than to SFW, both on SD and ESD.
> > > > >
> > > > > By contrast, the 2D t-SNE plots are purely for visualization, and it is well known that t-SNE can distort global geometry and make distributions that are separable in the original space appear partially overlapping in the embedding [1].
> > > > >
> > > > > The high-dimensional measurements in W2&Q7 and Q2 reflect the true geometry of the representations. The separations in the original feature space are in fact more pronounced than what a low-dimensional t-SNE projection can convey. Our detector is trained on the full features (not on t-SNE) and its high accuracy demonstrates that the underlying representations remain discriminative even when the 2D t-SNE visualization looks less clearly separated.
> > > > >
> > > > > [1] Wang, Yingfan, et al. "Understanding how dimension reduction tools work: an empirical approach to deciphering t-SNE, UMAP, TriMAP, and PaCMAP for data visualization." Journal of Machine Learning Research 22.201 (2021): 1-73.
> > > > >
> > > > >
> > > > > ***F2: Further Explanation for Q1&Q4.***
> > > > >
> > > > > As noted in our responses to Q1&Q4, we have re-run the experiments on adversarial NSFW prompts using the widely adopted Stable Diffusion Safety Checker. We consistently use Safety Checker to determine whether a generated image is harmful rather than relying on NudeNet. For adversarial NSFW prompts, the actual presence of harmful content in the output is verified by Safety Checker and further quantified by the NSFW rates reported in W5.
> > > > > Importantly, concept-erasing baselines (ESD and SLD) only offers nudity checkpoints, so they cannot be applied to the naïve NSFW prompts.
> > > > >
> > > > >
> > > > > ***F3: Question about Metric Calculation.***
> > > > >
> > > > > We include the entire dataset in the calculation. We have explicitly evaluated the NSFW generation rates in W5 of our rebuttal. The majority of prompts (84.37% in average) across all categories consistently produce NSFW outputs, confirming that the datasets used for evaluation are sufficiently valid and representative.
> > > > >
> > > > >
> > > > > ***F5: Clarification on the Use of Concept-Erasing Methods.***
> > > > >
> > > > > Our IGD is designed as an in-generation NSFW detection mechanism, while concept-erasing methods such as ESD/SLD are generation-side defenses that aim to partially remove certain concepts from the model. They do not provide any NSFW detection capability. They only reduce the probability that some NSFW content is produced and their intentions and tasks are completely different. We just follow GuardT2I and treat ESD/SLD purely as concept-erasing methods that can, in a limited sense, hinder NSFW content generation.
> > > > >
> > > > > The results in Q6 for “Ours(ESD)” are meant only to show that IGD can be ported to an ESD backbone by reusing the same pipeline on its intermediate noise features, without changing our architecture. We are not intended to define or optimize a specific “ESD + IGD” combined defense, which is outside the scope of this work. In the revised version, we will clarify that (1) IGD is a detection method, (2) ESD/SLD are concept-erasing baselines, and (3) we only follow GuardT2I in comparing against concept-erasing defenses, but the intention and task are completely different.

---

### Official Review · Reviewer_oAzj · 2025-10-31

**Soundness:** 3
**Presentation:** 3
**Contribution:** 2
**Rating:** 4
**Confidence:** 4

**Summary:**

The paper introduces In-Generation Detection (IGD) for identifying NSFW content during the diffusion process itself.
To achieve this, the authors train a classifier on the noise predictions of a pre-trained diffusion model, distinguishing between NSFW and safe-for-training (SFT) samples.
This approach bridges the trade-off between speed (typical of pre-generation filtering) and accuracy (typical of post-generation detection).
Experimental results demonstrate that IGD significantly outperforms prior pre-generation methods, achieving stronger robustness against both naïve and adversarial NSFW content.

**Strengths:**

1. Clarity - The paper is well written and clearly structured.

2. Conceptual Simplicity - The method is conceptually simple yet effective.

3. Empirical Advantage - It substantially outperforms prior pre-generation NSFW detection methods.

4. Efficiency: Compared to post-generation detection approaches, IGD achieves conceptually faster detection though not shown.

**Weaknesses:**

1. Limited Novelty:
The research contribution is somewhat limited in scope. While the proposed approach offers potential speed advantages by detecting NSFW content. Classifiers on noisy intermediate representations are not new.
a. Given that the approach involves training a classifier on predicted noise, it would be interesting to explore whether this classifier could also be used as a guidance signal to steer generation away from NSFW regions, potentially broadening the impact of the method.

2. Incomplete Comparison with Post-Generation Methods:
The paper does not include a direct comparison with post-generation NSFW detection techniques, leaving open questions about actual efficiency and performance trade-offs:
a. What is the measured gain in inference time relative to standard post-generation detection?
b. If one were to perform partial denoising (e.g., generating a pseudo-image with only 3–5 denoising steps), could post-generation detection achieve comparable accuracy and speed to the proposed in-generation method?

**Questions:**

See previous section 1a. 2a. 2b.

**Details Of Ethics Concerns:**

The paper is using known datasets and open source and common diffusion models, no new ethics concerns.

---

> ### Author Response · Authors · 2025-11-22
>
> We sincerely thank you for the constructive suggestions! Below we respond to the comments in ***Weaknesses (W) and Questions (Q)***.
>
> ***W1: Limited Novelty.***
>
> We sincerely thank you for your suggesions.
> As you pointed out, classifiers on intermediate noisy representations have been explored in prior work, but our contribution lies in leveraging predicted noise within the in-generation process of T2I models, which removes the need to generate any visual-level content and significantly reduces safety-check latency.
>
> We also appreciate the reviewer’s suggestion on extending the classifier to act as a guidance signal to steer denoising trajectories away from NSFW regions. This is indeed a promising direction, and we will explore it in future work.
>
> ***W2: Inference Time of Post-detection.***
>
> To address the reviewer’s concerns regarding post-generation detection efficiency, we additionally compare our method with a commonly used image-level NSFW detection method Safety Checker. The results are listed below.
>
> It is important to clarify that both our method and the Safety Checker require the denoising steps of the T2I pipeline. Therefore, the major time cost in both methods comes from waiting for the T2I model to generate the predicted noise or image, rather than the classifier itself.
>
> Our in-generation method detects NSFW signals before image's generation, avoiding any image decoding and reducing the cost to 0.0044 s in inference mode comparing to Safety Checker's 0.0171 s.
>
> | Method                      | Avg Time per Sample (s) | GPU Memory (MiB) |
> |-----------------------------|---------------------------|-------------------|
> | Ours                        | 0.5304                    | 626               |
> | Ours (inference time)           | 0.0044                    | 626               |
> | Safety Checker              | 5.2777                    | 1872              |
> | Safety Checker (inference time) | 0.0171                    | 1872              |
>
> The two tables below report Accuracy, and the results show that IGD achieves the best or near-best performance across almost all benchmarks.
>
> | Method          | I2P-sexual | MMA-Diffusion | Ring-A-Bell | Sneaky-Prompt | P4D   | DIFFZOO |
> |-----------------|------------|---------------|-------------|----------------|-------|---------|
> | Safety Checker  | 65.00%     | 61.25%        | 96.52%      | 57.00%         | 74.50% | 72.99%  |
> | Ours            | 94.25%     | 94.00%        | 94.78%      | 93.25%         | 93.00% | 93.32%  |
>
> | Method         | sexual | violence | self-harm | harassment | hate    | shocking | illegal_activity |
> |----------------|--------|----------|-----------|------------|---------|----------|------------------|
> | Safety Checker | 63.50% | 51.00%   | 57.00%    | 57.50%     | 59.57%  | 53.00%   | 51.00%           |
> | Ours           | 89.50% | 89.00%   | 89.50%    | 90.50%     | 95.74%  | 90.00%   | 89.00%           |
>
>
> ***W3: Performances on Partial Denoising of Post-detection.***
>
> We further evaluated the reviewer’s suggestion:
> decode intermediate diffusion latents (e.g., 3–5 steps) into pseudo-images via VAE and feed them into the Safety Checker.
>
> However, our experiments show that: Safety Checker classified all pseudo-images (no matter SFW or NSFW) decoded from early diffusion latents as NSFW, regardless of the actual semantic content at that stage. As a result, the output becomes uninformative and cannot be used for meaningful detection.

---

### Note · Authors · 2025-12-01

**Comment:**

We sincerely thank all ICLR PCs, ACs, and reviewers for your time, effort, and valuable feedback on our paper.

**Withdrawal Confirmation:**

I have read and agree with the venue's withdrawal policy on behalf of myself and my co-authors.